# SARS-CoV-2 antibodies recognize 23 distinct epitopic sites on the receptor binding domain

Jiansheng Jiang [1✉], Christopher T. Boughter [2], Javeed Ahmad[1], Kannan Natarajan [1], Lisa F. Boyd[1], Martin Meier-Schellersheim [2] & David H. Margulies [1✉]

The COVID-19 pandemic and SARS-CoV-2 variants have dramatically illustrated the need for a better understanding of antigen (epitope)-antibody (paratope) interactions. To gain insight into the immunogenic characteristics of epitopic sites (ES), we systematically investigated the structures of 340 Abs and 83 nanobodies (Nbs) complexed with the Receptor Binding Domain (RBD) of the SARS-CoV-2 spike protein. We identified 23 distinct ES on the RBD surface and determined the frequencies of amino acid usage in the corresponding CDR paratopes. We describe a clustering method for analysis of ES similarities that reveals binding motifs of the paratopes and that provides insights for vaccine design and therapies for SARS-CoV-2, as well as a broader understanding of the structural basis of Ab-protein antigen (Ag) interactions.

[1] Molecular Biology Section, Laboratory of Immune System Biology, National Institute of Allergy and Infectious Diseases, NIH, Bethesda, MD 20892, USA.
[2] Computational Biology Section, Laboratory of Immune System Biology, National Institute of Allergy and Infectious Diseases, NIH, Bethesda, MD 20892, USA.
✉email: jiangji@niaid.nih.gov; dmargulies@niaid.nih.gov

Our ability to predict protein interactions is still very limited despite great progress in the application of computational methods for determining protein structures from amino acid sequence alone[1,2]. This limitation is even more evident with regard to the interactions among highly variable immune receptor surfaces as dictated by Ab complementarity determining region (CDR) loops and the antigenic structures they bind. Accordingly, efforts directed toward providing systematic analyses or rational design strategies for Ab-Ag interactions need to incorporate experimentally determined structural data on specific Abs. Recent efforts in Ab design take advantage of segmental approaches[3] or extensive computational resources[4,5]. Such hindrances emphasize the importance of incorporating as much information on naturally occurring specific Ab-Ag structures as possible. Here, we report a systematic structural analysis, taking advantage of the thousands of structures of SARS-CoV-2-derived proteins, including spike and various Ab complexes that have been determined to further our understanding of the fundamental mechanisms of the pathogenesis and neutralization of SARS-CoV-2 in the context of the human immune system. Many Abs have been reported to have potent neutralizing activity, preventing spike interaction with the cellular receptor, angiotensin converting enzyme (ACE) 2. Several Abs have been developed as therapeutics and have variable efficacy against variants of concern (VOC). Our analysis of available structures may aid in understanding which Abs may be of value for emerging variants and contribute to evolving strategies for prophylaxis, treatment, and immunization.

Ab-protein antigen (Ab-Ag) interfaces have been a focus of immunologists and protein chemists for more than 80 years[6], not only because of the important role of Abs in defense against infection[7], but also due to the general interest in understanding protein-protein interactions[8]. High resolution structural analysis of protein-protein complexes, based initially on X-ray crystallography and more recently on cryogenic electron microscopy (cryo-EM), provides an objective basis for understanding not only the biophysical principles that determine affinity and specificity, but also for elucidating biological and evolutionary rules that govern immunological molecular recognition of foreign molecules and pathogens[9,10]. With an ever-expanding database of detailed Ab-Ag structures, great attention has been directed to the characterization of such molecular interfaces, particularly as an understanding of the rules of engagement might permit rationalization of the reactivity of existing Abs, the design of Abs with new binding activities, and strategies for design of immunogens that might elicit more broadly neutralizing Abs[11–13].

The widespread infectivity, variance, and molecular characterization of the SARS-CoV-2 virus have provided a wealth of information concerning the functional and structural biology of the immune response. At the beginning of the SARS-CoV-2 pandemic, many laboratories accomplished detailed structural characterization of anti-RBD Abs and nanobodies (Nbs, single domain antibodies), leading to a classification of Abs based on the location of their footprints on the RBD surface. Initially, four classes of Ab were categorized, based on the orientation of the RBD bound and whether the Ab blocks infectivity or binding to the cellular receptor, ACE2[14] (Supplementary Table 1). A receptor binding motif (RBM) has been defined as those RBD residues that specifically interact with ACE2[15]. Binding analysis of Nbs and human mAbs derived from patients along with a limited number of protein structures assigned five surface regions of the RBD reflecting its antigenic anatomy[16]. Epitopic analysis was further extended by the definition of seven "communities" of Abs that bind to the RBD surface[17]. Recent analysis of anti-RBD Ab and Nb as well as molecular dynamics analysis in the context of evolving escape mutations has taken advantage of these earlier

classification schemes[18–23]. Others have analyzed a number of anti-spike Nb in terms of their affinity and neutralization capacity[24]. The functional classification of RBD epitopes, i.e. those that block infectivity of SARS-CoV-2, is valuable in identifying Abs likely to be of immediate therapeutic benefit during a rapidly spreading pandemic. The structure-based, function-agnostic, approach described here captures a broader set of RBD epitopes and is aimed primarily towards understanding the physico-chemical basis of epitope-paratope interactions. Such an understanding can enable predictions of antibody reactivities of new RBD variants based solely on RBD amino acid sequences.

Although these classification schemes have been valuable and adopted widely in the analysis of Abs as to how they bind to RBD and spike, particular Abs and Nbs may not be unambiguously classified (Supplementary Fig. 1). The previous summaries were based on a relatively small number of available structures and focused on the relative superposition of the Abs in the complexes, rather than on a comparison of the epitopic contacts of the RBD surface. In particular, the original distinction between Class 1 and Class 2 seemed clear based on the initial structures. However, as more structural models became available, apparent inconsistencies arose. For example, Ahmad et al.[25] determined that synthetic Nbs Sb16 and Sb45 contacted both Class 1 and Class 2 epitopic surfaces and approached the RBD from different angles. As more structures of Ab and Nb complexes are determined, it is apparent that an expansion of the initial classification scheme is warranted.

In this work, we focus on complexes of Abs and Nbs bound to the RBD of the spike protein to generate a comprehensive structural framework to further our understanding of Ab- and Nb-RBD recognition. Using a large database, we offer a structure-based classification exploiting quantitatively defined contacting amino acid residues on the RBD as well as a clustering analysis. These analyses reveal common characteristics of some 23 frequently contacted ES and the structural nature of the surfaces of the RBD that interact with Ab/Nb. We also systematically analyze the molecular features that define these antibodies and, by applying a rigorous evaluation of the surface features of the RBD that are seen by Abs and Nbs, generate general insights into the fundamental nature of Ab-Ag recognition. This analysis should facilitate the characterization of new anti-RBD antibodies as they arise.

## Results

**Identification of epitopic sites (ES).** To identify common features of ES of the RBD, we systematically investigated structures of Abs (as Fabs and Fvs, Ab fragments that confer antigen binding activity) and of Nbs (as VHH or synthetic library-derived sybodies) in complex with the spike protein or its RBD as collected in the CovAbDab[26] and the protein data bank (PDB)[27,28]. Abs and Nbs that bind the SARS-CoV-2 RBD are summarized in Table 1. As of 12/22/2022, a total of 6746 Ab and 620 Nb sequences have been collected in the CovAbDab. Of the Abs, 6321 are human, including those from vaccinees, and 390 derive from humanized mouse or phage display Ab libraries. For Nbs, 620 sequences derive from camelids (alpaca/camel/llama), of which 276 are from camelid-derived phage display libraries, some naïve, some immunized. Among these sequences, structural coordinates for only ~5% of the Abs and ~10% of Nbs were available in the PDB, and we compiled a non-redundant list of 340 Ab and of 83 Nb X-ray or cryo-EM structures (Supplementary Data 1 & 2) which serve as the basis of our structural analysis.

Evaluation of the biophysical properties that contribute to protein-protein interactions may be based on different criteria, including calculation of free energy terms of interacting residues[29], measurement of shape complementarity (Sc[30]), and calculation of buried or accessible surface area[31–35]. We elected to simplify this analysis first by calculating interatomic contacts

**Table 1 Summary of sequences and structures of anti-SARS-CoV-2 antibodies and nanobodies.**

| Origin/Source | Sequences | Structures * (PDB IDs) |
|---|---|---|
| Total antibodies | 6746 | 340 |
| Human (patient or vaccinee) | 6321 | 250 |
| Mouse (immunized/humanized) | 165 | 40 |
| Phage display library or engineered | 225 | 33 |
| Undefined | 35 | 25 |
| Total nanobodies | 620 | 83 |
| Immunized (alpaca/camel/llama[#]) | 332 (123/18/189) | 50 (16/5/27) |
| Phage display library | 276 | 23 |
| Undefined | 14 | 12 |

The sequences and origin/source are collected in CovAbDab[26], as of 12/20/2022. The number of structures of antibodies and nanobodies in complex with RBD or spike protein are downloaded from PDB.
*Unique non-redundant structures determined either by X-ray or cryo-EM as listed in the PDB.
[#]This includes two sequences/structures from mice engineered to express llama Nb genes[77].

between Ab (paratopic) and Ag (epitopic) residues at the interface because the biophysical basis of binding (due to charge, hydrophobicity, hydrogen bonding and van der Waals interactions) is reflected in such contacts. For the hydrogen bond interactions, the distance is usually cut-off about 3.8–4.0 angstrom, but for non-bonded interaction or van der Waals interaction, it could be up to 5–8 angstroms. Generally, distances within 4–6 angstroms (Å) are considered indicative of direct contacts between interacting proteins. For computational approaches cut-offs that rang from 5 to 10 Å due to the dynamics feature of proteins may be used. Vangone and Bonvin[36] studied the correlation of the contact distance and the binding affinity, and found the approximate distance range is between 4.0 to 5.5 Å. Viloria et al.[37], determined an optimal distance cut-off for contact-based protein networks of 5.0 Å. Krawzyk et al.[38] used 4.5 Å to define epitope contacts. We adopted the contact distance cut-off at 5.0 Å for our Ab-Ag interaction, based on comparison of different cut-offs from 4.0 to 5.5 Å (see Methods). We plotted the numbers of Ab (paratope) contacts as hits versus the residue number of the RBD (epitope) for the Ab heavy (H) (Fig. 1a) and light (L) (Supplementary Fig. 2a) chains individually, and also overall for both H and L chains together (Supplementary Fig. 2b). We also plot the number of hits of the 83 Nbs to each RBD residue (Fig. 1c). For 340 Abs, H chains contribute 5623 contacts and L chains 3107 (Supplementary Table 2). By comparison, for 83 Nbs, 1836 contacts are observed. Thus, the number of contacts is ~25 per Ab and ~22 per Nb. Although the RBD residues bound by either Ab, H chain, or Nb are by and large, the same, the relative distribution of hits varies for several regions. In particular, the region from RBD residue 368 to 386 is recognized more frequently by Nbs, while other contiguous surfaces are seen equivalently (Fig. 1a, c). The numbers of hits for Ab H chains are represented graphically as a heat map on the RBD surface in Fig. 1b, and the heat maps for the Nbs are shown in Fig. 1d.

Several contiguous stretches of amino acids of the RBD that make Ab contact were apparent, although the frequency of hits varied considerably for different regions on the surface of the RBD. A fine-grained tabulation of regions of the RBD consisting of three to nine residues define each individual ES as shown in Table 2. Each of these ES may be assigned to either of the four major classes identified earlier or to the RBM recognized by the ACE2 receptor (Table 3). These regions include distinct secondary structural features such as strands, loops, turns, and helices (Supplementary Movie 1), and represent contacts seen by few (<0.3%) to many (>10%) Abs. Consideration of the secondary structural features (loops, turns, or short β strands) and the accessible surface area prompts the identification of 23 distinct contiguous sites, including regions

encompassing residues 404 to 421 that had been overlooked in previous studies. The hit numbers are not evenly distributed over the RBD surface, and it is difficult to distinguish which binding sites belong to the previously defined Class 1 or Class 2 due to overlaps generated by the reduction of the three-dimensional surface to a two-dimensional plot. Figure 2a, b displays these ES on the RBD surface with the ES numbers for Abs (magenta) and Nbs (blue) respectively. The thickness of the putty cartoon indicates greater hit numbers. The computed accessible surface area (ASA) (see Methods) for each individual ES (Table 2) ranged from ~100 Å$^2$ to more than 500 Å$^2$. The total buried surface area (BSA) was also computed for each of 340 Abs (for H chain, L chain and H plus L chain) and for 83 Nbs as in Supplementary Data 1 (for Abs) and 2 (for Nbs). The values of BSA range from 64 Å$^2$ to 1112 Å$^2$ for Ab H, from 0 Å$^2$ to 912 Å$^2$ for Ab L, and between 264 and 1824 Å$^2$ for H plus L of the 340 Abs. BSA for the 83 Nb ranges from 437 Å$^2$ to 1412 Å$^2$.

As an indication of the relative immunogenicity of each of the 23 ES, we tabulated the proportion of Abs and Nbs that recognized each site (Fig. 2c). Approximately 7 to 11% of Ab H chains recognized ES11, 13, 16, 18, and 20, which represent ES contained within the previously defined Class 1 and Class 2 regions. In general, Nb recognition of specific ES was similar to that of Ab H chains, with the predominant recognition representing from 7 to about 10% of Nbs see Table 2 and Fig. 2c, falling within Class 2 and Class 4. Notable differences in the predominant ES recognized by Abs and Nbs are that ES8, 13, 16, and 18 are more frequently seen by Abs while ES4, 5, 6, 7, 11, and 20 are more frequently identified by Nbs. For example, ES16 was recognized by 10% of Abs and by 0.16% of Nbs. This difference may be explained since ES16 forms a solvent exposed convex structure which may not be conducive to recognition by Nbs. By contrast, ES4, 5, and 6 form a contiguous patch, recognized more frequently by Nbs, a region that is not exposed to solvent in the complete spike when the RBD is in the down position. Thus, Nbs may be better able to access such hidden surfaces, perhaps because of their relatively small size (12kD compared to ~25 or 50 kD for Fv and Fab respectively or ~150 kD for complete bivalent IgG, with corresponding three-dimensional volumes)[39]. Alternatively, since many Nbs were identified based on binding to isolated RBD, some epitopes identified from such screens may be partially hidden in the complete spike protein. In comparing L chains with H chains, as shown in Fig. 2d, L chains generally contribute less to these ES. Nevertheless, L chains seem to preferentially contact ES7, 20 and 21. We note that some ES (e.g. ES7, 8, 9, and 23) could not be placed into the previous classification schemes and some sites overlap on Class 1 and Class 2 (i.e. ES12, 19, and 20). However, most of the 23 ES may be

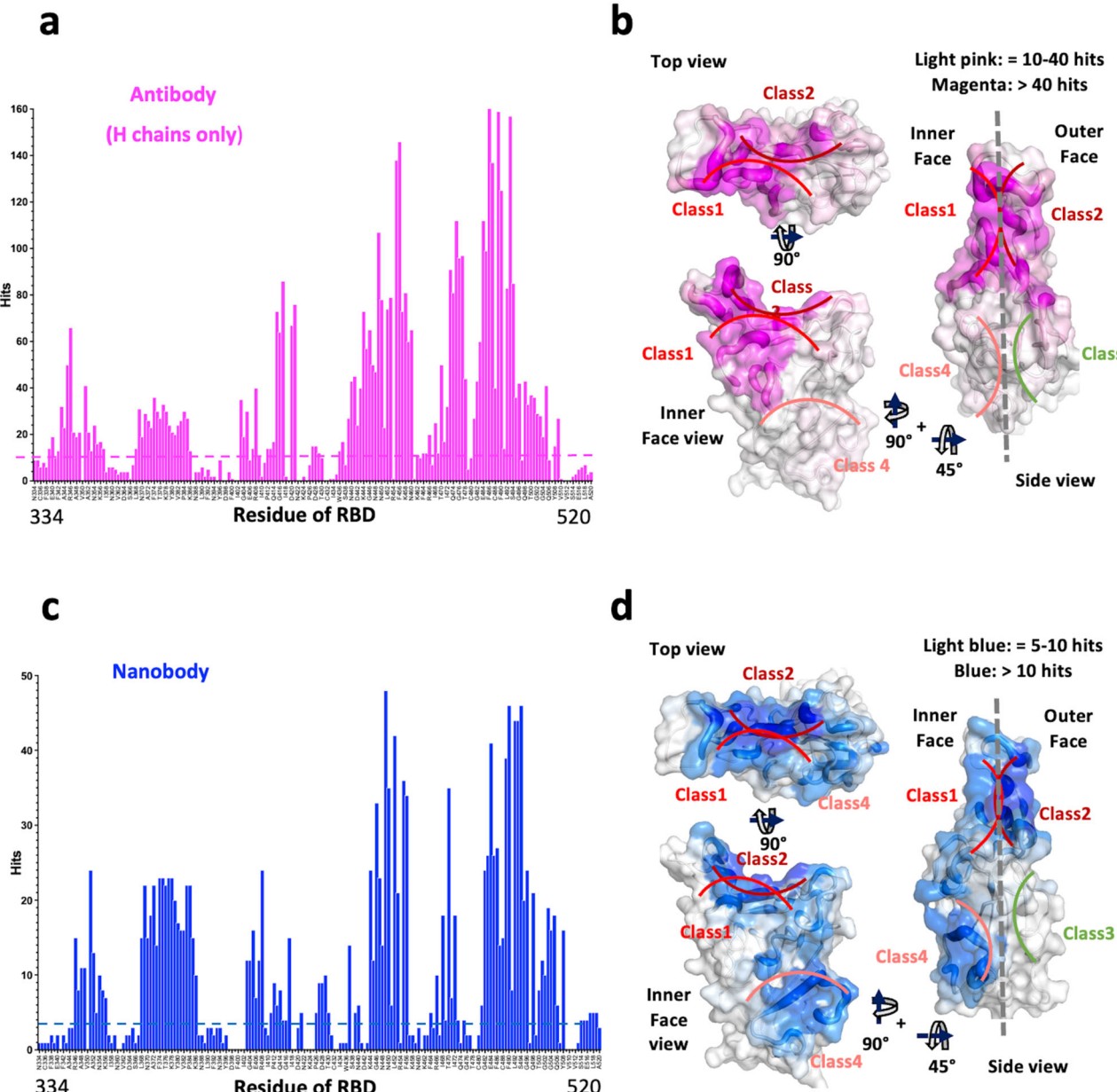

**Fig. 1 Number of contacts to RBD by Abs and Nbs. a** Total number of contacts to each of the indicated RBD residues summed from all available X-ray and cryo-EM structures from Ab H chains. The odd numbered RDB residue names are omitted for clarity. **b** Graphic depiction of number of contacts illustrated as footprint on the RBD and as putty heat map of RBD cartoon backbone. Top, inner face, and side views of RBD are shown. **c** Total number of contacts as in (**a**), but for Nb contacts. **d** Surface footprint and putty heat map of Nb contacts as in (**b**).

viewed within the four classes described by Barnes (Table 3)[14]. In addition, the RBM of the RBD[15] may be defined in terms of the ES that overlap the ACE2-RBD interface (i.e. ES8, 11, 12, 13, 16, 18, 19, 20, 21, and 22 (Table 3)). With these 23 fine-grained ES, we extend the prior classification for Class 1 to now include ES8 and 9 (Table 3). Each ES surface area or footprint is illustrated by a color map of the RBD surface (Fig. 2e, Supplementary Movie 2). The sum of these 23 ES covers as much as 70% of the total accessible surface area (ASA) of the isolated RBD, illustrating the breadth of the human antibody response to RBD.

**Analysis of CDR loop contributions and epitope-paratope interactions.** The CDRs in the hypervariable region of Abs play critical roles in recognizing antigens[9,40,41], and their variability in sequence and length facilitates interaction with distinct

antigenic epitopes[42]. We tabulated the number of contacts for each CDR loop or non-CDR residues of 340 H chains and L chains and 83 Nbs to each of the 23 ES. The contact percentages are summarized in Fig. 3a, b, c respectively. The corresponding statistics are listed in Supplementary Table 2a, b, c. For Ab H chains (Fig. 3a), CDR loops account for 82% of the contacts to ES (CDR1 = 16%, CDR2 = 21%, CDR3 = 45%), while only 18% of the contacts are from non-CDR residues. Interestingly, CDR1 of H chains plays a major role in binding to ES16. For Ab L chains (Fig. 3b), CDR1 loops play a major role (40%) in binding to RBD while CDR3 represent only 25% of the contacts. One explanation for the reduced the role of the CDR3 loop of L chains might be that their average length (10 aa for 340 Abs) is generally shorter than that of H chain CDR3 (15 aa for 340 Abs), see Fig. 3d. For Nbs (Fig. 3c), CDR represent 73%

**Table 2 Definitions of epitopic sites (ES) seen by Abs and Nbs.**

| ES | RBD residue (range) | Amino acid sequence | Structural feature | ASA (Å$^2$) | Abs (H chain) (%) | Nbs (%) |
|---|---|---|---|---|---|---|
| 1 | 339–341 | GEV | α-helix | 155 | 0.78 | 0.16 |
| 2 | 343–349 | NATRFAS | loop | 455 | 4.07 | 2.83 |
| 3 | 351–357 | YAWNRKR | 3$_{10}$ ->β-strand | 410 | 2.60 | 3.76 |
| 4 | 368–374 | LYNSASF | α-helix->loop | 469 | 3.17 | 7.03 |
| 5 | 375–380 | STFKCY | β-strand | 287 | 2.90 | 6.97 |
| 6 | 381–386 | GVSPTK | Loop->3$_{10}$ | 381 | 2.49 | 5.50 |
| 7 | 403–409 | RGDEVRQ | 3$_{10}$ | 321 | 2.76 | 4.68 |
| 8 | 411–417 | APGQTGK | loop | 342 | 4.91 | 2.78 |
| 9 | 420–428 | DYNYKLPDD | α-helix->loop | 371 | 3.24 | 1.69 |
| 10 | 437–443 | NSNNLDS | β-strand->α-helix | 378 | 3.59 | 1.47 |
| 11 | 444–449 | KVGGNY | loop/strand | 457 | 7.10 | 8.39 |
| 12 | 450–454 | NYLYR | β-strand | 194 | 4.57 | 5.72 |
| 13 | 455–460 | LFRKSN | loop | 448 | 9.98 | 4.25 |
| 14 | 462–467 | KPFERD | loop | 470 | 1.28 | 1.31 |
| 15 | 468–472 | ISTEI | loop | 384 | 2.44 | 4.47 |
| 16 | 473–479 | YQAGSTP | β-strand->loop | 466 | 9.34 | 0.71 |
| 17 | 481–484 | NGVE | loop | 404 | 4.36 | 5.28 |
| 18 | 485–487 | GFN | loop | 275 | 7.06 | 3.65 |
| 19 | 488–491 | CYFP | β-strand->loop | 181 | 5.98 | 5.88 |
| 20 | 492–496 | LQSYG | β-strand | 126 | 7.15 | 9.69 |
| 21 | 497–502 | FQPTNG | loop | 307 | 3.38 | 2.94 |
| 22 | 503–509 | VGYQPYR | 3$_{10}$->β-strand | 253 | 2.42 | 4.14 |
| 23 | 516–520 | ELLHA | loop | 548 | 0.44 | 1.20 |

RBD residue range for each ES is indicated, along with the amino acid sequence, secondary structural features (as determined by DSSP[78]), accessible surface area (ASA) (see Methods) of the contacting residues, and percentage of Ab H chains and Nbs.

**Table 3 Correlation of ES with class definitions.**

| Class | Epitopic sites (ES) |
|---|---|
| 1 | 7, 8, 9, 13, 16, 18, 21, 22 |
| 2 | 10, 11, 12, 15, 17, 19, 20 |
| 3 | 1, 2, 3, 14, 23 |
| 4 | 4, 5, 6 |
| RBM | 8, 11, 12, 13, 16, 18, 19, 20, 21, 22 |

correlation of ES with Class definitions by Barnes[14] and with receptor binding motif (RBM)[15]. Those ES that represent greater than 7% of contacts by Ab H chain or Nb are indicated in boldface.

(CDR1 = 13%, CDR2 = 14%, CDR3 = 46%) of the contacts to the RBD surface, while 27% involve non-CDR residues. The average length of Nb CDR3 is 16 aa. Thus, for both Ab H chains and Nbs, CDR3 contributes the greater proportion of those residues that interact with the RBD, reflecting a major role for CDR3 in RBD recognition. (Illustrations of CDR1, CDR2, and CDR3 contacts are shown in Fig. 5c).

We plotted the frequency of particular amino acids used by Abs and Nbs (paratopic residues) that interact with particular ES of the RBD for Ab H chains (Fig. 4a) and for Nbs (Fig. 4b). These are shown as heat maps. The residues listed on the top of the panel represent the most frequently contacting amino acids for the specific ES. The frequency of usage of each amino acid for Abs (pink) and Nbs (blue) is compared in Fig. 4c. Tyrosine (Y), serine (S), and arginine (R) are the three amino acids most preferred for binding any ES of RBD (Fig. 4c). Previous analyses of paratopic preferences for a wide range of Abs recognized a high frequency of tyrosine usage[43]. We also observed that tryptophan is more frequently used in Nbs as compared with Abs (Fig. 4c). The usage of CDR3, CDR2 and CDR1 amino acids is plotted in Fig. 4d, e, f respectively. To illustrate the predominance of particular paratopic residues of the Ab H chains that contact specific ES, we also grouped these as WebLogo plots[44] (Supplementary Fig. 3).

**Cluster analysis of epitopic sites and binding motifs**. Having identified the sets of ES bound by each Ab and Nb (see Supplementary Data 1, 2), we then grouped the Abs and Nbs by computation of the similarity of the ES recognized (see Methods). Similarity of a pair of ES sets is a value between 0 and 1 reflecting recognition of completely different (0) or identical (1) sites. This clustering method compares ES sets on the RBD without visualization of graphic models. Assigning a similarity threshold of 0.85 (see Methods) results in the identification of 33 clusters for Abs, designated A1 to A33 (Supplementary Table 3a) and 10 clusters for Nbs, N1 to N10 (Supplementary Table 3b). Although Abs within a single cluster bind the same subset of ES, they may, or may not address the RBD from the same angle or utilize CDR of the same length or composition. These differences are illustrated in Fig. 5a for clusters A1, A3, and A11 for H chains and in Fig. 5b for clusters N1, N3, and N4 for Nbs. The members of nanobody cluster N4 reveal a similar orientation because they have the same conformation and length of CDR loops. Abs or Nbs within the same cluster recognize the same contiguous RBD surface and are expected to compete sterically.

CDR loops contain sequence motifs for epitope recognition[45–48]. To identify such motifs we analyzed a subset of interfaces from cluster A1, designated A1S1, that recognized ES with a similarity of ≥0.9. A1S1 consists of 28 members (cluster A1 has 56 members of similarity ≥0.85). All the members of A1S1 recognize the same ES set (ES8, 9, 12, 13, 16, 18, and 19) (Fig. 5c), utilize the same CDR loops, and superpose well. Analysis of the residues of CDR1, 2, and 3 that contact the RBD indicated those residues that are preferentially utilized by this stringently selected cluster of Abs. For the binding motifs of CDR1, 2, and 3 of A1S1, the favored residues are summarized in a WebLogo plot (Fig. 5c). Remarkably, Y, S, G, and T predominate for all CDR except CDR3 which exploits R in most instances. Thus, application of a more stringent ES similarity score helps to identify the preferred binding motif utilized by the Ab of the same subgroup. This stringent grouping of Abs and Nbs, based on high similarity score of their respective ES, may prove a useful adjunct in structure

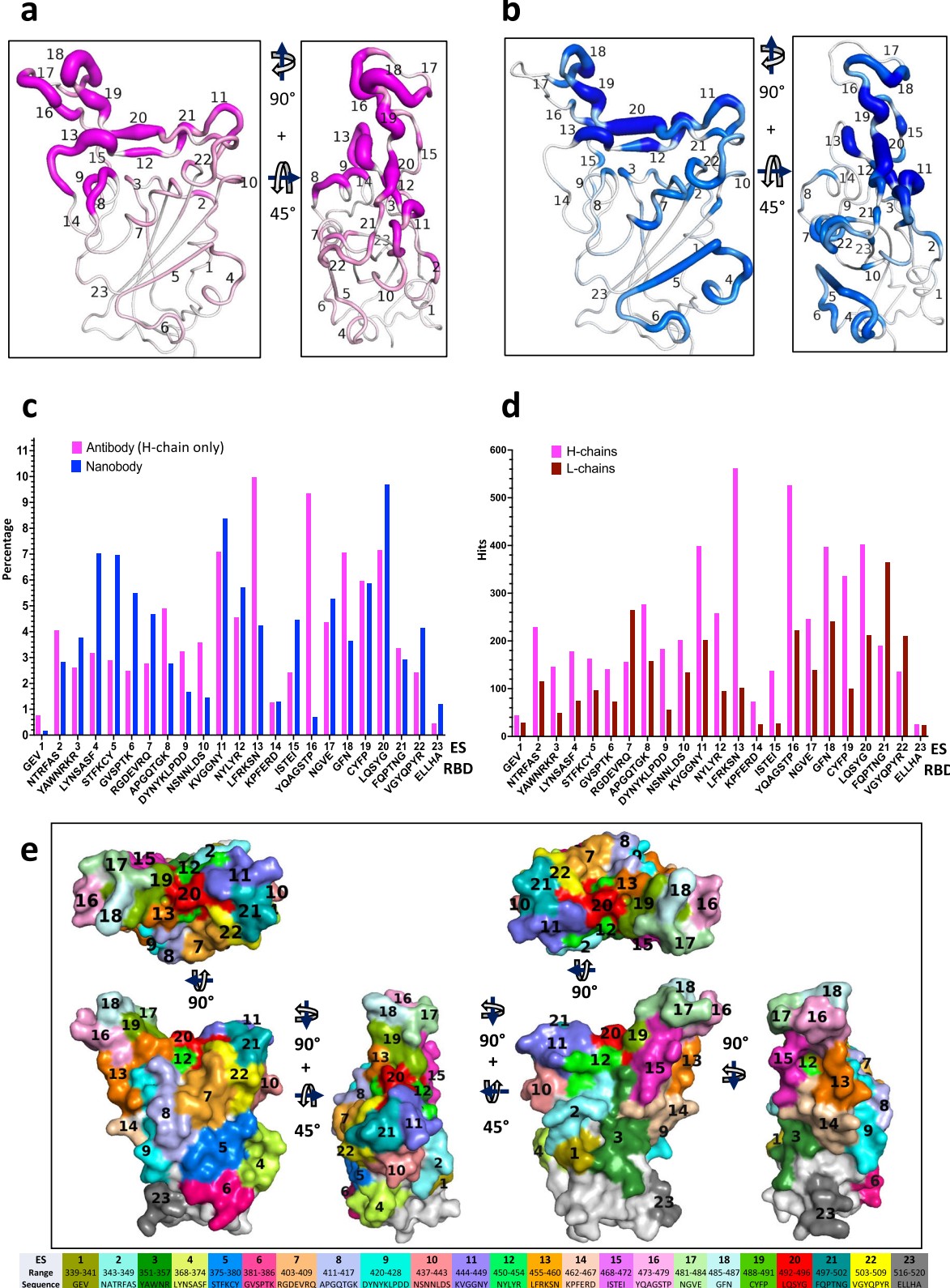

**Fig. 2 Distribution of Abs and Nbs on RBD surface. a** Putty heat map of H chain of antibody with the definition of ES. The thickness of putty represents the number of hits. **b** Putty heat map of Nb with the definition of ES. **c** Distribution of Abs/Nbs on ES of RBD surface (percentage, %). Magenta represents Ab, blue represents Nb. **d** Comparison of antibody H chains and L chains on ES of RBD surface (by hit number). **e** ES surface area or footprint is illustrated by a color map of the RBD surface. A movie is presented in Supplementary Movie 1.

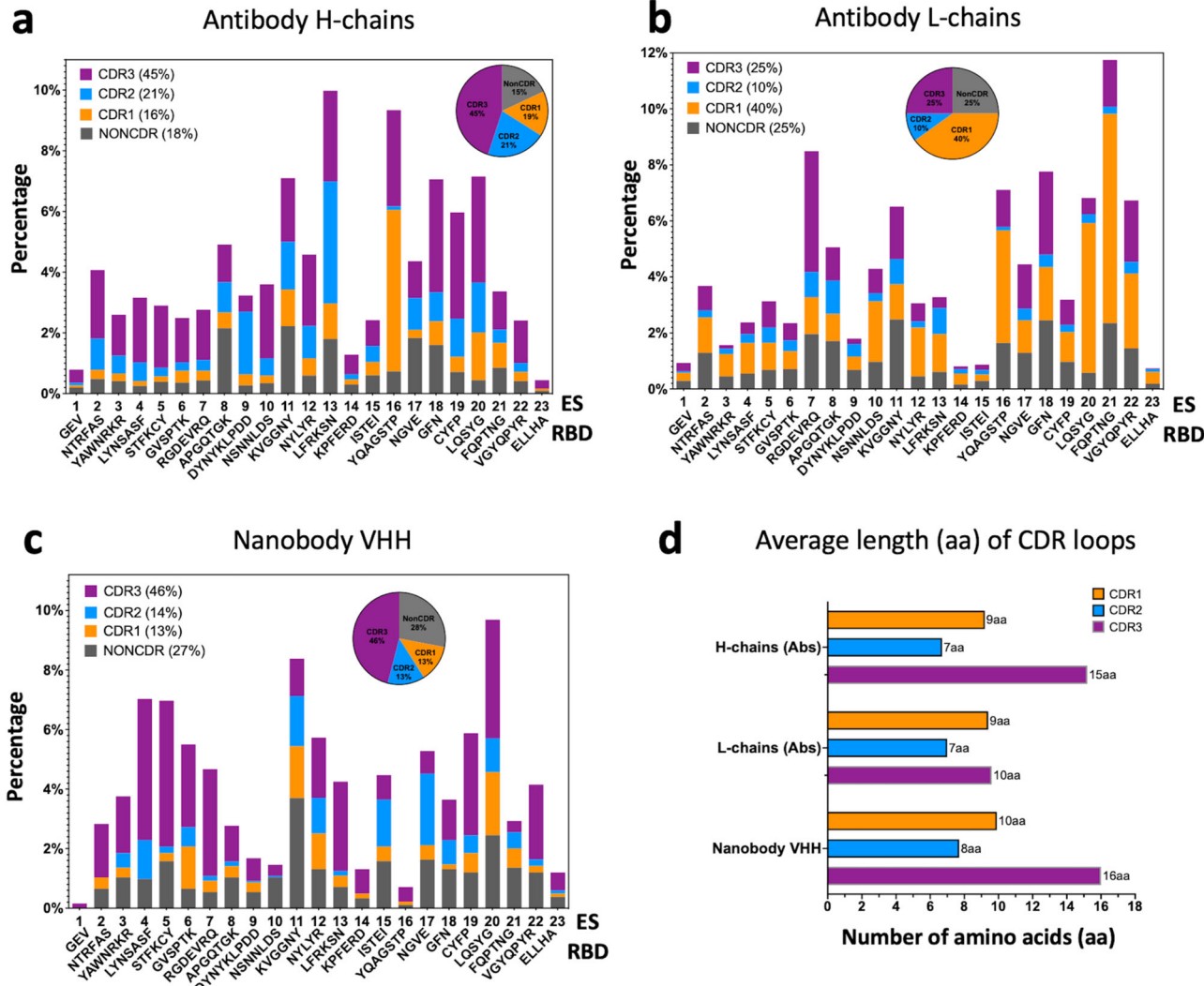

**Fig. 3 Distribution of CDR loops of contacts to RBD surface over ES. a** Antibody H chains are plotted (percentage). Pie graph indicates the composition of CDR1 (16%, orange), CDR2 (21%, marine blue), CDR3 (45%, purple) and non-CDR (18%, gray) respectively. **b** Antibody L chains are plotted (percentage). Pie graph indicates the composition of CDR1 (40%, orange), CDR2 (10%, marine blue), CDR3 (25%, purple) and non-CDR (25%, gray) respectively. **c** Nanobody chains are plotted (percentage). Pie graph indicates the composition of CDR1 (13%, orange), CDR2 (14%, marine blue), CDR3 (46%, purple) and non-CDR (27%, gray) respectively. **d** Average length (in amino acids (aa)) of CDR loops extracted from the sequences (CovAbDab[26], as of 12/20/2022) and used in this study. The averages are over 340 antibodies and 83 nanobodies respectively.

prediction based on amino acid sequence and antibody competition. Previous work identified the over-represented public class of mAbs encoded by IGHV3-53 and IGHV3-66 that neutralize the spike[45,49,50]. We also investigated the V(D)J gene combinations representing those mAb structures (Supplementary Fig. 6a). Among 6316 mAb sequences in the CovAbDab, the top three IGHV genes are 3–30, 1–69 and 3–53, and IGHJ genes are 4, 6 and 3. However, the top IGHV genes for the structural representatives are 3–53 and 3–58 and IGHJ gene 4, 6, and 3 combined (see blue heat map). A large cluster based on the gene combination similarity, GA1 (IGHV3-53/IGHJ6), as shown in Supplementary Fig. 6b, has an ES set of (8,9,13,16,18,19) which is related to the cluster of A1S1 (Fig. 5c). However, GA1 is a subset of the cluster A1S1 (17 vs 28 members).

To extend the utility of our ES definitions, we set out to determine broad biophysical trends common among the Abs that cluster to each ES region. Using the automated immune molecule separator (AIMS) software[51], a tool which characterizes immune molecules without structural knowledge, we analyzed similar SARS-CoV-2-specific Abs. With this we identified 11 clusters which are designated

as AIMS1, AIMS2, etc (Fig. 5d). Not all Abs in a single AIMS cluster bind the same ES. However, AIMS6 and AIMS7 overlap as subsets of cluster A1 and have a similarity score of 0.85.

**Relation of ES and SARS-CoV-2 escape mutations**. SARS-CoV-2 variants have evolved rapidly from Alpha, Beta, Delta, and Omicron with multiple mutations and deletions. The development of the latest Omicron subvariants can be traced from BA.1, BA.1.1, BA.2, BA.3, BA.4/5, and XBB.1 to XBB.1.5 and they incorporate as many as 30 mutations and deletions in their RBDs[52–54]. Table 4 lists the mutations in these variants and the ES to which they map. Subvariants marked "X" have different substitutions at a given position. Table 5 lists the major Omicron subvariants and their associated ES. (For example, XBB.1.5 has substitutions of P and S for V445 and G446, respectively, which are contained in ES11, and substitution of S and Q for F490 and R493, respectively, which are in ES19). Similarly, XBB.4 preserves the same substitutions, but also substitutes R for L452 in ES12. Figure 6a–d illustrates the location of these variants on the RBD

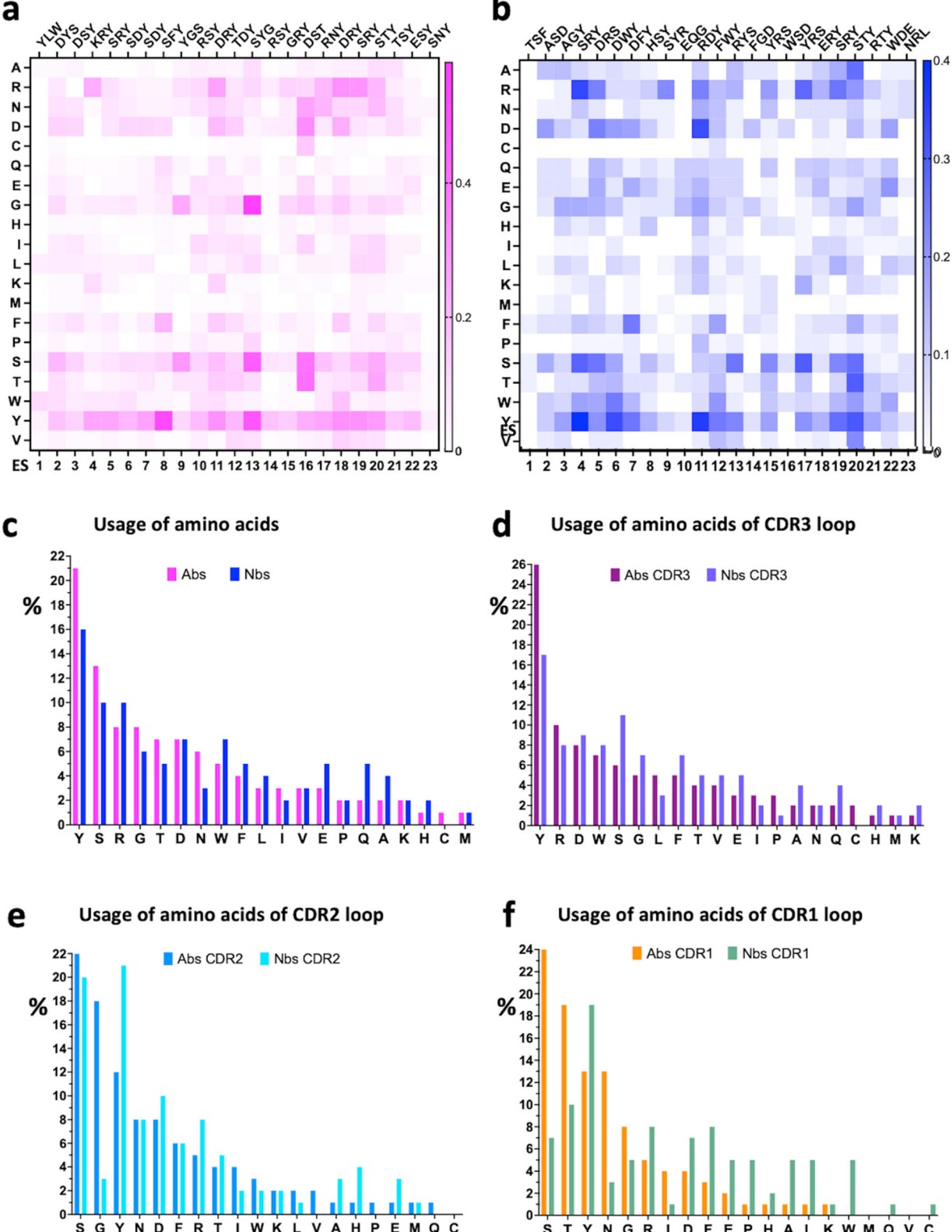

**Fig. 4 Distribution of amino acids of Abs/Nbs over ES. a** Heat map of amino acids of Ab H chains on each ES, magenta indicates the frequency of the amino acids. **b** Heat map of amino acids of nanobody on each ES, blue indicates the frequency of the amino acids. Top triplets are those most frequently observed amino acids of Abs/Nbs on each ES. **c** The usage of amino acids of antibody H chains (magenta) and nanobodies (blue) in interacting with RBD is plotted in descending order (percentage). YSR are most frequently observed amino acids both for Ab and Nb. **d** Usage of amino acids in CDR3 loops (purple for Abs; light blue for Nbs). W of Nbs has relatively higher percentage in comparison to Abs both overall and for CDR3 loop. **e** Usage of amino acids in CDR2 loops (marine for Ab; cyan for Nb). SGY are most frequently used amino acids for Ab, SYD are most frequently used amino acids for Nb. **f** Usage of amino acids in CDR1 loops (orange for Ab; green for Nb). STYN are most frequently used amino acids for Ab, TY are most frequently used amino acids for Nb.

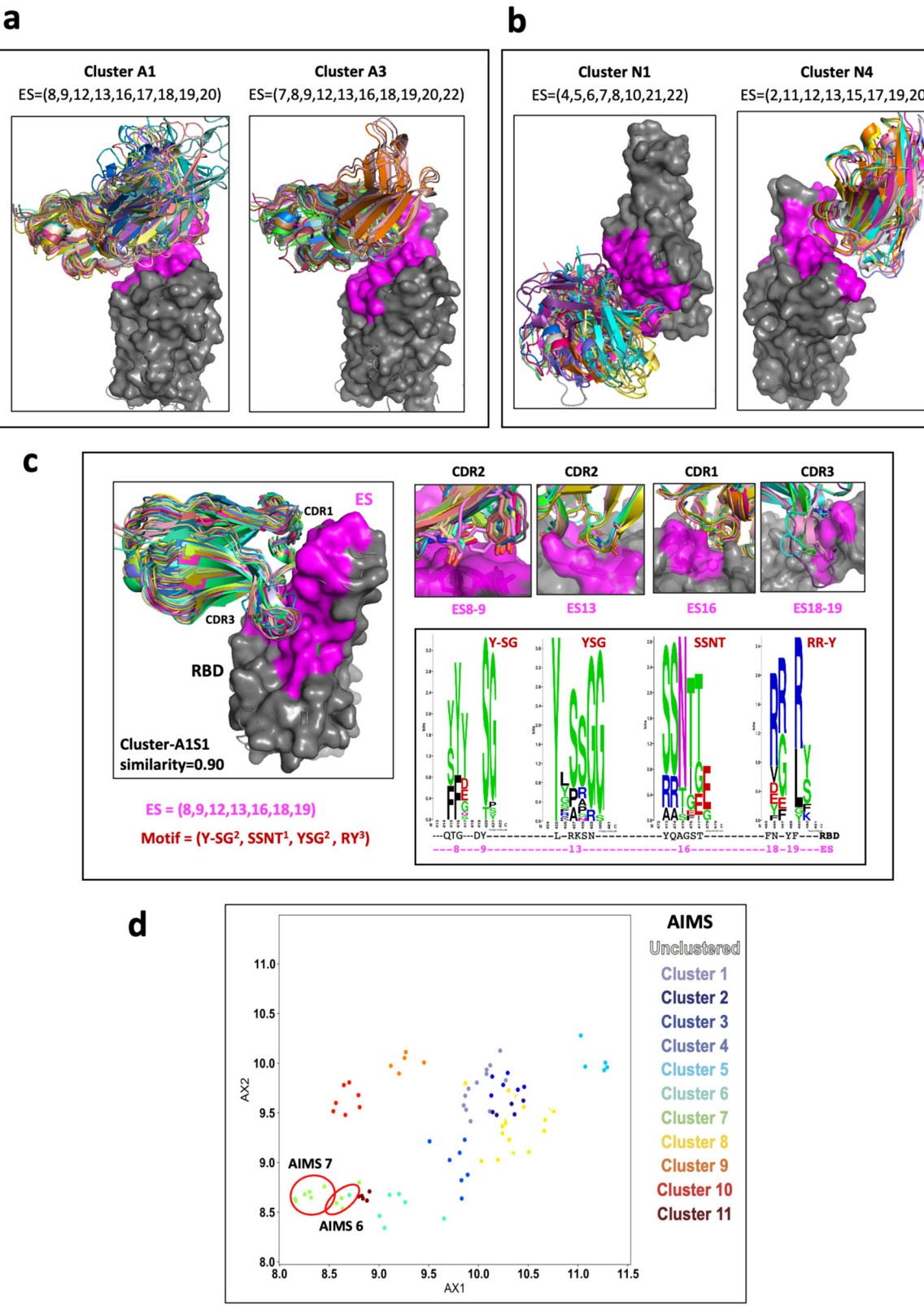

surface for Omicron and their mutation sites are matched to one or more of the 23 ES. Strikingly, Omicron escape mutations are distributed throughout several distinct ES of the RBD (Table 4, Fig. 6a–d), posing a formidable challenge in the design of new vaccines and therapeutic antibodies. Notably, mutations in ES3, 6,

9, 14, 15, and 23 have not yet been reported. These ES for which mutations have not yet been reported are illustrated in Fig. 6e.

Our comprehensive analysis of RBD epitopes and their corresponding Ab paratopes offers the possibility of identifying currently approved SARS-CoV-2 therapeutic Abs that may be

**Fig. 5 Identity of the similarity of the ES and clustering of Abs/Nbs (see Supplementary Table 3). a** Illustration of three antibody clusters: A1 and A3, each identifies a specific ES combination. Superimposed are members of the cluster on the RBD (only HV domains are shown for clarity). **b** Illustration of three nanobody clusters: N1 and N4. RBD is presented as gray surface, magenta indicates the binding areas (footprints) of ES of RBD. **c** A subset of the Cluster A1, named A1S1, ES = (8,9,13,16,18,19) with similarity ≥ 0.90, shows a strong binding motif on CDR loops. The members (28) of A1S1 are superposed on the RBD on the left panel. On the right top panel are shown the contacts between CDR loops and the binding sites (ES8-9, ES13, ES16, and ES18-19). On the right below panel, WebLogo plots show the amino acids from Abs binding to ES8-9, 13, and ES18-19 respectively. Y,S,G from CDR2 are favor binding to ES13 (RBD residues from 455–460); S,S,N,T of CDR1 favor binding to ES16 (RBD residues 455–459); and R and Y of CDR3 are favor binding to ES18-19 (RBD residues 485-491). **d** Clustering using AIMS[51]. Here AX1 and AX2 are "principle components" of biophysical properties, or "mature information". AIMS6 and AIMS7 are very similar to A1S1.

**Table 4 Relation of ES to SARS2-CoV-2 escape mutations.**

| Variant | Alpha | Beta | Delta | Omicron | Sub (X) | ES |
|---------|-------|------|-------|---------|---------|-----|
| Mutations | | | | | | |
| | | | | G339X | D,H | 1 |
| | | | | R346X | T,K,E,I,S | 2 |
| | | | | S371L | | 4 |
| | | | | S375F | | 4 |
| | | | | T376A | | 5 |
| | | | | D405N | | 7 |
| | K417N | K417N | K417N* | K417X | N,T,D | 8 |
| | | | | N440K | | 10 |
| | | | | L441K | | 10 |
| | | | | K444X | T,R,N,M,G | 11 |
| | | | | V445X | A,P,N | 11 |
| | | | | G446X | D,S,Y | 11 |
| | | | L452R | L452X | Q,R,M | 12 |
| | | | | F456L** | | 13 |
| | | | | N460X | K,S,Y | 13 |
| | | | | S477N | | 16 |
| | | | T478K | T478X | K,R | 16 |
| | E484K* | E484K | | E484A | | 17 |
| | | | | F486X | I,V,P,S,A | 18 |
| | | | | Y489L | | 19 |
| | | | | F490X | I,L,S,V,C,R,Q | 19 |
| | | | | R493X | Q,L,P | 19 |
| | S494P* | | | | | 20 |
| | | | | G496S | | 20 |
| | | | | Q498R | | 21 |
| | N501Y | N501Y | | N501Y | | 21 |
| | | | | Y505H | | 22 |

Major mutations with the main lineage of variants of SERS-Cov-2 and corresponding ES site. "X" column indicates the amino acid substitutions of the subvariants of Omicron. (*mutations found in some but not all indicated variants, i.e. nonsignature mutations; **Latest mutation of Omicron, as July 2023, EG.5 = XBB.1.9.2.5, it appears in ES13).

used to neutralize emerging SARS-CoV-2 variants and Omicron subvariants. The latest reported structures[47,55–57] describe some Abs that bind these subvariants. We can identify a number of Abs or Nbs that target particular ES sets that are either mutated or preserved in emerging variants. Those Abs/Nbs exhibiting multiple contacts to contiguous ES sites with concomitantly large buried surface area and high binding affinity deserve the greatest attention. Thus, using Ab/Nb structures already determined that target particular ES, we can model the effects of the variant mutations on antibody recognition.

Two examples illustrate this approach: the R346T RBD mutation in the subvariants BA.4, BA.5, BF.7 and XBB.1.5 lies within ES2 (Table 2, Table 4, Fig. 6d), and those Abs that recognize ES2 may be further evaluated for their ability to bind the mutants that harbor the R- > T substitution. Supplementary Table 4a lists a number of Abs and Nbs whose structures are known that interact with ES2, and analysis of several Abs which may potentially resist the escape mutation (Supplementary Fig. 4a). Specifically, the emergency use authorized (EUA) mAb S309 (one of three Fab modeled in PDB 7JX3) (sotrovimab) may

have neutralizing potency when combined with other antibodies to BA.1.1.529, BA.1, BA.2.75 subvariants[58,59]. A second is the F486 mutation found in XBB.1 (F486S) and XBB1.5 (F486P) which is located in ES18 and 19 (F490 & R493). We identified a number of Abs and Nbs (Supplementary Table 4b) that have multiple contacts with ES17, 18, and 19, such as for COVOX-45, which preserves those to P486 from the main-chain of the CDR3 loop. Also, the nanobody Nb-2-67 makes multiple hydrogen bonds to maintain contact with ES18 (Supplementary Fig. 4b).

Our analysis of ES recognized by Abs and Nbs and the identification of specific ES affected by mutations in VOC provides an explanation for the ineffectiveness of some Ab that have been tested therapeutically. One example, Evushield™, which consists of two Abs, tixagevimab (AZD 8895) and cligavimab (AZD 1061) illustrates this point. These Ab have been studied by X-ray crystallography (tixagevimab, PDB 7L7D, both tixagevimab and cligavimab in 7L7E[60]) and by cryo-EM[61]. By our analysis, tixagevimab interacts with ES13, 16, 18, 19, and 20 and cligavimab with ES2, 10, 11, and 12. As shown in Table 4, residues in every one of these ES are mutated in the Omicron variant. This then explains the lack of beneficial effect of Evushield™ and supports a molecular basis for the recent revision of its EUA by the FDA (https://www.fda.gov/drugs/drug-safety-and-availability/fda-announces-evusheld-not-currently-authorized-emergency-use-us). In particular, Omicron variant XBB.1.5 (see Table 3b), harbors mutations that reduce efficient recognition by the Ab products (bamlanivimab plus etesevimab, casirivimab plus imdevimab, sotrovimab, and bebtelovimab), which are no longer authorized for use in the United States https://www.covid19treatmentguidelines.nih.gov/therapies/antivirals-including-antibody-products/anti-sars-cov-2-monoclonal-antibodies/#:~:text=Four%20anti%2DSARS%2DCoV%2D,mild%20to%20moderate%20COVID%2D19.

## Discussion

The enormous world-wide effort to elucidate the mechanistic underpinnings of the immune response to SARS-CoV-2 has provided deep insight into aspects of the B cell and T cell responses to infection and immunization and has contributed to ongoing strategies for therapy and prevention. Here, we have taken advantage of the ever-increasing structural database of anti-SARS-CoV-2 Abs and Nbs to analyze the three-dimensional features that are described by X-ray and cryo-EM structures of Ab and Nb complexes with the RBD of the virus, either alone or in the context of the full spike protein. We have developed several analytical computational tools described in detail in the methods that allow the tabulation and analysis of molecular contacts and ES between the Abs/Nbs and the RBD. These provide a convenient avenue for querying and comparing the binding sites and interactions of particular Abs/Nbs and will support additional queries as the CovAbDab and PDB entries increase. This has permitted the categorization of the epitope-paratope interactions and molecular surface characteristics that lend themselves to recognition by Abs and the recurrent structural motifs of the CDR residues of the Abs/Nbs. This identification of 23 ES derives

**Table 5 Latest mutations in the major (PANGO) lineage of subvariants of Omicron and corresponding ES site.**

| ES | ES2 | ES11 | | | ES12 | ES13 | ES18 | ES19 | |
|---|---|---|---|---|---|---|---|---|---|
| Mutations | R346X | K444X | V445X | G446X | L452X | N460X | F486X | F490X | R493 |
| B.1.1.529 | | | | S | | | | | Q |
| BA.2.13 | | | | | M | | | | R |
| BA.2.75 | | | | S | | K | | | Q |
| BA.3 | | | | S | | | | | R |
| BA.4.6 | T | | | | R | | V | | Q |
| BA.5.1.26 | T | | | | R | | V | | Q |
| BA.5.2 | | | | | R | | V | | Q |
| BF.7 | T | | | | R | | V | | Q |
| BQ.1 | | T | | | R | K | V | | Q |
| BQ.1.1 | T | T | | | R | K | V | | Q |
| XBB.1 | T | | P | S | | K | S | S | Q |
| XBB.1.5 | T | | P | S | | K | P | S | Q |
| XBB.4 | T | | P | S | R | K | S | S | Q |

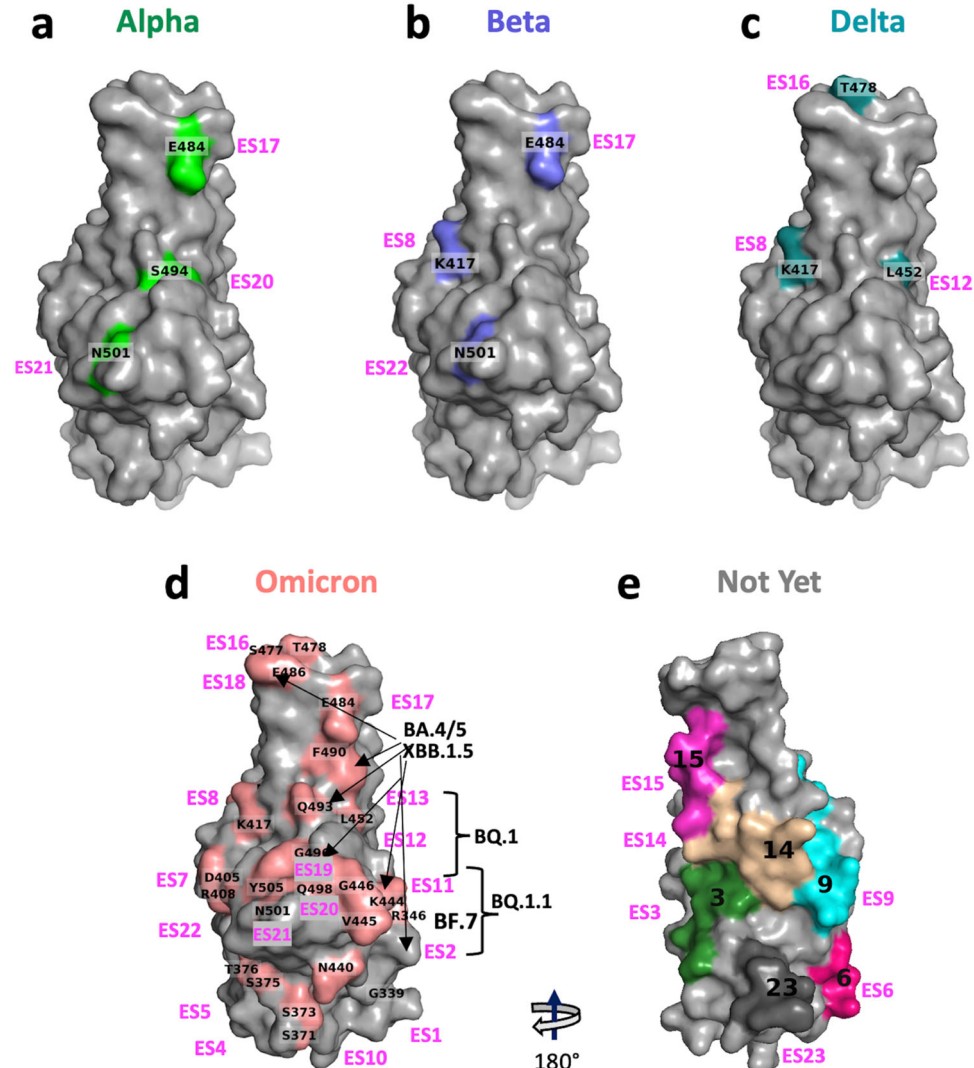

**Fig. 6 Ilustration of location of variant mutations and associated ES on RBD surface. a** Alpha variants. **b** Beta variants. **c** Delta variants. **d** Omicron variants and subvariants. **e** The ES areas where mutations have not yet been reported (color code refers to Fig. 2e).

from evaluation of a large number of Ab/Nb-RBD and Ab/Nb-spike structures and their interface contacts, and thus surpasses analyses based on amino acid sequence or gross structural comparison alone. Our method of clustering ES sites with various

stringencies, and independently of the antibodies that recognize them, offers an additional tool towards the goal of prediction of CDR sequences that recognize particular epitopic sites. This analysis is focused on the RBD alone, and does not take into

account potential contacts with the glycan moiety linked at N343. Only 27 of 340 Ab make any contact with residue N343.

Of some 340 Abs and 83 Nbs, our analysis indicates that the 23 ES on the RBD characterized in part by secondary structural features may be recognized at different frequencies. This fine-grained analysis of the RBD surface reveals that as many as 10% of Abs may recognize common features such as those of ES16 as seen by Abs, or of ES11 as seen by Nbs. Our findings and definition of 23 ES are not dependent on the distance cut-off parameter in computing the Ag-Ab contacts. Although the total numbers of contacts may increase slightly with a longer distance cut-off, the percentage of contacts in each ES bin are almost the same (as shown in Supplementary Fig. 5) with the distance cut-offs at 4.0 Å (gray), 4.5 Å (blue), 5.0 Å (red) and 5.5 Å (green) respectively (H chain only).

Understanding the biophysical or structural characteristics of antigenic or immunogenic sites on protein antigens has been a subject of considerable interest for many years, beginning with efforts to understand common sites seen by heterogeneous Abs and further refined as monoclonal Abs have been studied[6,40,42,43,62]. Recent efforts have identified common motifs that human antibodies exploit to bind similar epitopes[63]. Consistent features of antigenic sites include hydrophobicity, accessibility, and segmental mobility as well as sequence dissimilarity to the Ab-producing organism (tolerance). Here we have taken the opportunity to investigate a large number of Abs and Nbs for which the antigenic site of a single protein is defined at high resolution by structural criteria. Here we took advantage of currently available structures of complexes from a database of antibodies that derive largely from patients or vaccinees but also from mice and includes single chain antibodies (nanobodies) derived from immune or naïve libraries (see Table 1). Of course, analysis of structures compiled in any antibody database may be biased by a variety of factors including the biological source(s) of the antibodies (from natural infection or immunization; or from naïve or immune based libraries), whether they could be engineered effectively to produce adequate amounts of protein for X-ray or cryo-EM analysis, whether the proteins crystallized well, or how well they bind variant viral proteins. Despite such potentially confounding factors, several important consistent conclusions may be drawn: (1) common sites are recognized by both Abs or Nbs; (2) several major surfaces of the RBD have not been addressed by either Abs or Nbs; and (3) some sites are favored by either Abs (e.g, ES16 and ES18) or by Nbs (e.g., ES4 and ES5). This latter phenomenon may reflect germline VH gene preferences in the human (as suggested[64]) or the well-recognized characteristic of Nbs, whose relatively long CDR3 loops are capable of exploring concave surfaces[65].

Our analysis suggests that several regions of the RBD, that are recognized by a higher proportion of Ab may be particularly important to incorporate into peptide-based immunogens (such as ES11, 13, 16, 18, and 20) and that further generation vaccines might pay particular attention to new viral variants that affect these sites. Alternatively, Ab therapies may benefit from a focus on those reagents that recognize both common antigenic sites as well as those that are rarely identified. Although our analysis here has been confined to Abs/Nbs that recognize the RBD of the spike protein of SARS-CoV-2, this approach may, in principle, be applied to a variety of Abs/Nbs directed against proteins of pathogenic organisms.

## Methods

**Datasets.** Covid Ab and Nb sequences were culled from the Coronavirus Antibody Database, CovAbDab (http://opig.stats.ox.ac.uk/webapps/covabdab/)[26] and coordinates of three-dimensional models were taken from the protein data bank (PDB) (https://www.rcsb.org/;

and https://rcsb.org/covid19/)[27,28]. Using the CovAbDab list as of 12/20/2022, we downloaded all complexes of Ab/spike or Ab/RBD structures determined by X-ray crystallography or cryo-EM from the PDB. The total number of downloaded PDB entries is about 595 (see the "Related Structures" column in Supplementary Data 1). We manually curates this structure dataset as follows: (a) We removed the structures with resolution worse than 5.0 Å; (b) when the same Ab appeared in two or more different structures, we selected the one of highest resolution; (c) if an Ab had both X-ray and cryo-EM structures, we removed the one that was of worse resolution, regardless of the method; (d) since the spike is a trimer, if two or more of the same Ab bound to different chains of the same spike, we considered only one complex with this Ab; (e) in the case of bispecific or bivalent Ab or multiple Ab/Nb in the same PDB structure, we evaluated two or more such "unique" Ab/RBD complexes. (f) CDR designations were taken from the CoVAbDab database, which follows the IMGT numbering system. The sequence designations of CDRH3 and CDRL3 were taken directly from the CoVAbDab download. CDRH1, CDRH2, CDRL1, CDRL2 were identified by sequence alignment.

This procedure resulted in 340 unique "non-redundant" neutralizing antibodies from 595 PDB entries (see Supplementary Data 1). We performed the same analysis for Nb (from a list of 124 structures reduced to 83 "non-redundant" Nb, see Supplementary Data 2). We did not rectify any missing residues or atoms and kept the original model from the downloaded PDB structures. We used the CNS 1.3 script, "contact.inp" with the distance cut-off at 5.0 angstroms to compute the interacting contacts between Abs and RBD for these 340 structures. If two or more residues from an Ab contacted identical residues on the RBD within 5.0 angstroms, we list them as multiple contacts. This procedure produced the "raw" epitope-paratope contact dataset (see github.com/jiangj-niaid/RBD-SARS2/340absH-contact-dis-0207.txt) for the following analysis step. Of the curated structures, 59 of the 340 Ab and 4 of 83 Nb structures were determined with variant RBDs. Our analysis included all variants, but eliminating those variants from the analysis showed little difference in the residue/hit plots. (See Suppplementary Data 1 and 2).

Buried surface area (BSA) is an important structural character and a quantitative measurement of interaction at the interface. BSA values correlate to the sum of individual contacts and directly link to the binding affinity or neutralizing potency. We also calculate the BSA values (using PISA/CCP4 program) for each chain (H and L) of an antibody/nanobody.

Based on these curations and evaluations, we created the annotation files for each Abs and Nbs; see Supplementary Data 1 and Data 2. [the Data 1 file contains the following data items: Name of Abs, PDB id, epitope-chain-id, paratope-chain-ids, R/S (RBD alone or spike), X/E (X-ray or cryo-EM), Resolution, BSA H chain, BSA L chain, BSA H + L chains, ES H chain, ES L chain, Related Structures, Variants and list of Mutations (on RBD). The Data 2 file contains the following data items: Name of Nbs, PDB id, epitope-chain-id, paratope-chain-id, R/S (RBD alone or spike), X/E (X-ray or cryo-EM), Resolution, BSA, ES, Related Structures, Variants and list of Mutations (on RBD). (see github.com/jiangj-niaid/RBD-SARS2/)].

The produced "EPI contact datasets" composed of the following data items for each contact: PDB id and name of Abs or Nbs, Chain-ids (epitope-paratope), Residue-name of RBD, Residue-id of RBD, ES-id, Residue-name of Abs or Nbs, Residue-id of Abs or Nbs, Distance, CDR-id. (See github.com/jiangj-niaid/RBD-SARS2/).

**Software.** All analyses were performed with our EPI (Epitope-Paratope Interaction) software package of mixed scripts in C-

shell, perl, and python. EPI software is available at https://github.com/jiangj-niaid/EPI/. Supplementary Fig. 7 shows the flowchart of this package. We downloaded the sequences, including the extracted CDR sequences from CovAbDab, also downloaded all structures of Ab/Nb in complex with spike or RBD from the PDB and deduced a "non-redundant" structural dataset (see "Datasets" above). The Input files and parameters include: (a) A list of PDB IDs and names of Ab/Nb along with the epitope chain and paratope-chain dersignations. (b) Predefined ES residue range (in Table 2). (c) Predefined CDR loops. (d) A list of known variants and mutations. (e) The contact distance cut-off (default is 5.0 Å). The program will generate "EPI contact datasets" according to the input files and pdbid list. Separate sub-dataset for Ab H chain, L chain, or Nb may be designated. Based on this EPI dataset, the analysis scripts then perform a structural alignment and evaluate the statistics of the CDR loops and amino acid usage. Additional variants and mutants may be detected.

Contact distances were calculated based on scripts taken from CNS 1.3 (http://cns-online.org/v1.3/)[66]. Buried surface area (BSA)[34,67,68] was calculated with PISA (Proteins, Interfaces, Structures and Assemblies[34]), and accessible surface area (ASA)[35,69–72] was calculated with CNS 1.3.

The clustering method used in EPI is based on the ES (i.e. RBD binding sites) not amino acid sequences of Abs or Nbs. The numbers of ES (1–23) are then converted to a corresponding string of 23 letters from "a" to "w" and the similarity between sets of ES is computed using the Normalized Edit Distance that was developed from the Hamming[73] and Levenshtein[74] Distance. A similarity of 1 indicates that the two strings or two ES sets are identical; a similarity of 0 indicates that the two strings or ES sets are completely different. The similarity is then calculated for pairwise combinations of all Abs or Nbs based on their ES sets. Abs or Nbs can be clustered by imposing a similarity threshold. For 340 Abs we empirically evaluated similarity thresholds from 0.50 to 0.99 at 0.05 intervals and found that a similarity threshold of 0.85 yielded 33 clusters. For 83 Nbs a similarity threshold of 0.85 yielded 10 clusters.

The AIMS analysis package[51] used for biophysical clustering of antibody sequences can be found at https://github.com/ctboughter/AIMS, including generalized Jupyter Notebooks and a Python-based GUI for the replication of the results presented herein or for the application of this analysis to novel datasets. Detailed descriptions of the foundational concepts critical for this analysis and the instructions for use can be found at https://aims-doc.readthedocs.io.

Figures for structural models are generated by using PyMOL[75] (https://pymol.org/2/). Sequence logo figures were generated with WebLogo (https://weblogo.berkeley.edu/)[44]. Sequence alignments were made with Clustal Omega (https://www.ebi.ac.uk/Tools/msa/clustalo/)[76]. Graphic plots were generated with Prism 9 (https://GraphPad.com).

**Reporting summary**. Further information on research design is available in the Nature Portfolio Reporting Summary linked to this article.

## Data availability
All data generated for analysis in this study have been made available on ZENODO at https://zenodo.org/record/8241951 (ref: doi:10.5281/zenodo.8241951).

## Code availability
All code generated for analysis in this study with our EPI (Epitope-Paratope Interaction) software package of mixed scripts in C-shell, perl, and python have been made available on GitHub at https://github.com/jiangj-niaid/EPI/.

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

## Acknowledgements
This research was supported by the Intramural Research Program of the National Institute of Allergy and Infectious Diseases, NIH.

## Author contributions
J.J. conceived the project, wrote programs, analyzed and discussed data, prepared figures. C.T.B. contributed program scripts, analyzed and discussed data, and prepared figures. J.J., C.T.B., J.A., K.N., L.F.B., M.M.-S. and D.H.M. analyzed and discussed data, and wrote and revised the paper.

## Funding

## Competing interests
The authors declare no competing interests.
