## [Peer Review File · Communications Biology]

Reviewers' comments:

Reviewer #1 (Remarks to the Author):

Since the beginning of the COVID-19 pandemic, significant efforts have been dedicated to characterizing the structures of antibodies in complex with the SARS-CoV-2 spike protein. Various classification systems have emerged to categorize antibodies and epitopes. In this study, the authors examined 340 monoclonal antibodies (mAbs) and 83 nanobodies targeting the receptor binding domain (RBD) of the spike protein. They identified 23 distinct epitope sites on the RBD and analyzed the frequency of amino acid usage in the paratopes. The authors also established a correlation between epitope mutations and viral escape from antibody neutralization. This manuscript provides valuable insights into the understanding and classification of SARS-CoV-2 RBD epitopes. However, certain questions remain to be addressed.

Specific comments:

1. Page 4, lines 71-72 and page 5, lines 81-82: "Initially, four classes of Ab were categorized, based on the orientation of the RBD bound and whether the Ab blocks infectivity or binding to the cellular receptor, ACE2.... particular Abs and Nbs may not be unambiguously classified" As pointed out by the authors here, the Barnes classification of four classes of antibodies, classes 1-4, were not defined by the epitopes, but based on the orientation of RBD and whether the Ab competition with receptor binding. However, this classification system has been widely used to represent antibody epitopes, regardless of other factors, e.g. different angles of approach etc., which is one of the major reasons of the ambiguity, besides the "relatively small number of available structures" (line 83). Can the authors discuss more about the usage of the function-based classification system to represent epitopes?
2. In Figures 1a, 1c and supplementary Fig. 2, the bars are too narrow to distinguish and associate each bar with its respective residue.
3. In the structural analyses, did the authors take account of structures with SARS-CoV-2 variants? Or only structures with wild-type spikes/RBDs were analyzed?
4. In Figures 1a, 1c and supplementary Fig. 2, many residues are missing, e.g. 417, 484, 486.
5. Page 17, line 329 "Although non-random factors may contribute to biases in the available database". The authors may want to expand discussions about biases introduced by the structural database analyzed here. For example, some antibodies may be frequently elicited in immune system but difficult to isolate owing to the stabilized bait used, or the difficulty of determining structures of some major classes of antibodies targeting cryptic sites on SARS-CoV-2 spike.
6. Page 9, lines 151-152 "The values of BSA range from 106 Å² (PDB 6XDG) to 1112 Å² (PDB 7N64) for 340 Abs and from 444 Å² (PDB 7JVB) to 1412 Å² (7D2Z) for the 83 Nbs." Did the authors analyzed the heavy chains only for mAbs? If yes, can the authors also provide BSA numbers for heavy + light chains?
7. When calculating BSA values of RBDs by antibodies, did the authors calculate the interface made by the RBD-N343 glycan?
8. Which antibody numbering system (Kabat? IMGT?) did the authors use in this study to assign residue numbers and CDR lengths in this study?
9. CDRH1 plays an exceptionally major role in binding ES16, which may be related to the big cluster A1S1. Is this observation related to the over-represented public class of mAbs encoded by IGHV3-53/3-66 [1,2]?
10. Page 12, lines 216-218: "Assigning a similarity threshold of 0.85 (see Methods) results in the identification of 33 distinct, non-overlapping, clusters for Abs, designated A1 to A33 (Supplementary Table 4a) and 10 distinct clusters for Nbs, N1 to N10 (Supplementary Table 4b)." But the antibodies shown in Supplementary Table 4a seem overlapped. For example, CV30, CC12-3 are in both A1 and A2. CR3022, C1C-A3, Asarnow-3D11 are in both A11 and A17. Some antibodies like REGN10987 are not found in any of these clusters.

11. Table 6 cited in the main text should be Table 3?
12. In Figure 6 and Table 3, S494P and E484K are listed as Alpha mutations, and K417N in Delta. But normally these mutations are not considered as signature mutations in these variants. In addition, Some Omicron subvariants can contain G339H and T478R, in addition to G339D and T478K listed here.
13. Figure 6d: What are differences of pink and red residues? Why are some of the ESs highlighted?
14. Page 15, line 278: R486 should be F486.
15. Page 15, line 287: 7L7E contains both tixagevimab and cilgavimab.

References

- [1] Yuan, M., Liu, H., Wu, N. C., Lee, C. C. D., Zhu, X., Zhao, F., ... & Wilson, I. A. (2020). Structural basis of a shared antibody response to SARS-CoV-2. *Science*, 369(6507), 1119-1123.
- [2] Zhang, Q., Ju, B., Ge, J., Chan, J. F. W., Cheng, L., Wang, R., ... & Zhang, Z. (2021). Potent and protective IGHV3-53/3-66 public antibodies and their shared escape mutant on the spike of SARS-CoV-2. *Nature Communications*, 12(1), 4210.

Reviewer #2 (Remarks to the Author):

In this manuscript, the authors have utilized a computational clustering method and available crystallographic structures to identify 23 unique epitomic sites on the receptor binding domain (RBD) of the SARS-CoV-2 virus, the causative agent of the COVID-19 pandemic. This study is both intriguing and relevant, as it offers potential insights for the development of small molecules/proteins/peptides capable of neutralizing concerning variants that exhibit mutations in the RBD. Additionally, I appreciate the authors' examination of the distinctions between antibody (Abs) and nanobody (Nbs) epitope sites, a comparison that has received limited attention in prior research studies. Previous studies, one example listed here:

<https://www.frontiersin.org/articles/10.3389/fimmu.2021.691715/full> have focused on a much smaller subset of epitope sites, and as the author discuss in their introduction, not all Abs can be classified accurately within these "generic" classification schemes. I think the level of detail in 23 epitope sites and the 33 distant, non-overlapping clusters for Abs, will be useful in characterizing newly discovered Abs with less potential for immune escape, as well as potentially targeting the RBD with small molecule/peptide inhibitors at these epitope sites that are more resistant to immune escape. The analysis conducted by the author also offers insights into the immune escape of other commercial antibodies, including Evushield. This serves as a valuable example of how this research can provide a rationale for understanding immune escape mechanisms, particularly as new variants of SARS-CoV-2 continue to emerge. However, there are mistakes in the text, tables and figures. Therefore, I suggest that this paper be considered for publication in *Communications Biology* once the following issues have been addressed:

1. The authors need to check the text, figures, and tables as they don't match in many places.
 - a. Line 177, the authors say "each ES surface area or footprint is illustrated by a color map of the RBD surface (Figure 2d). I think the authors mean Figure 2e? Please check.
 - b. Line 242, I do not see Figure 5d in the files supplied for review? It is included in Fig.5 description but there is no corresponding 5d figure.
 - c. There is no Table 6a, maybe it is Table 3a
 - d. ES2 is not present in Figure 6a, maybe it is 6d
 - e. line 275: there is not supplementary figure 5a
 - f. Line 250 and line 252, authors list table 6a and table 6b. This is now table 3a and table 3b in their supplied files. Please fix throughout the manuscript where it is referred to as table
 - g. Line 275, there is no supp. Figure 5a? I think the authors mean Figure 4a? Same with line 282.

2. Can the authors include a little bit in the introduction about new omicron subvariants such as XBB.1.5 and how all FDA approved Abs are no longer effective.
3. How did the authors decide on a cut-off distance of 5.0 Å for their Ab-Ag interface contacts?
4. Table 3 (what authors have as Table 6 in the text) contains spelling mistake "SERS-cov-2". Please fix.
5. Line 259, authors state mutations in ES3,6,9,14,15, and 23 have not yet been reported. Could the authors create a separate figure with these epitopes highlighted and their corresponding residues? I think this will be helpful in showing where Abs or Nbs or other inhibitors could be targeted with less chance for immune escape by omicron.
6. Discussion: line 337, why do the authors recommend ES11, 13, 16, and 18 as sites to incorporate into peptide-based immunogens? Please provide a clarifying sentence based on your data why you recommend these sites.
7. Methods: line 368, please fix language "which users can make inquire". Maybe change to "in which users can inquire about a particular ES combination".
8. Figure 4d: This figure is only for CDR3; authors should also show figures for CDR1 and CDR2.
9. Line 191, in addition to Figure 3b, authors should also mention Figure 5C to show the CDR1, CDR2 and CDR3 regions.
10. Supplementary information: Authors should change "Excel files for Tables 2a and 2b are attached separately" to "CVS files (with names of the files) for Tables 2a and 2b are attached separately."
11. The article would have been more appealing if the authors had provided suggestions for antibodies, peptide inhibitors, mini proteins, protein mimics, or decoy receptors (such as sACE2) that could target a majority of the epitomic sites (from the 23 identified) in the currently circulating Omicron variants and demonstrate their ability to neutralize them through in vitro and in vivo efficacy data.

Reviewer #3 (Remarks to the Author):

The work of Jiang and colleagues deals with the investigation of the interaction between the receptor binding domain of SARS-CoV2 spike protein and a set of antibodies and nanobodies using structural data. In particular, studying the number of intermolecular contacts, the authors identified different epitopic sites on the RBD and cluster the antibodies in classes.

Overall the theme of understanding the interplay between antibodies and viral spike protein is important both from a theoretical and practical point of view. This work provides a detailed computational analysis, however, there are some points that in my opinion must be addressed before considering it for publication.

1) Presented results are obtained on the basis of the collected dataset, which comprises the most recent deposited structures. However, it is not clear to me what is the major progress with respect to other works already present in literature (see for instance Di Rienzo et al. "Dynamical changes of SARS-CoV-2 spike variants in the highly immunogenic regions impact the viral antibodies escaping", *Proteins: Structure, Function, and Bioinformatics*, <https://doi.org/10.1002/prot.26497> or Deshpande et al. 'Epitope Classification and RBD Binding Properties of Neutralizing Antibodies Against SARS-CoV-2 Variants of Concern. *Front Immunol.* 2021 Jun 4;12:691715. doi: 10.3389/fimmu.2021.691715). The authors should at least better discuss similarities and differences between their work and what is now present in literature.

2) SARS-CoV-2 spike protein is heavily glycosylated (see for instance Grant et al. "Analysis of the

SARS-CoV-2 spike protein glycan shield: implications for immune recognition." Sci Rep. 2020 Sep 14;10(1):14991.)

It is not clear if the authors consider this factor in their assessment of the epitopic sites.

3) Found results rely on a particular choice of cut-off distance (5 Å) for the definition of contacts and of the clustering method. Do the authors explore the robustness of their findings upon changing such parameters?

4) Please provide more details about the dataset. It is not clear how/if redundancy was taken into account, what resolution for the structure was considered, how missing residues were fixed, etc.

5) Please provide also the key details on the used algorithms (not only the reference to the software)

RESPONSE TO REVIEWERS

Reviewers' comments:

Reviewer #1 (Remarks to the Author):

Since the beginning of the COVID-19 pandemic, significant efforts have been dedicated to characterizing the structures of antibodies in complex with the SARS-CoV-2 spike protein. Various classification systems have emerged to categorize antibodies and epitopes. In this study, the authors examined 340 monoclonal antibodies (mAbs) and 83 nanobodies targeting the receptor binding domain (RBD) of the spike protein. They identified 23 distinct epitope sites on the RBD and analyzed the frequency of amino acid usage in the paratopes. The authors also established a correlation between epitope mutations and viral escape from antibody neutralization. This manuscript provides valuable insights into the understanding and classification of SARS-CoV-2 RBD epitopes.

We appreciate the Reviewer's recognition of the value of our work.

However, certain questions remain to be addressed.

Specific comments:

1. Page 4, lines 71-72 and page 5, lines 81-82: "Initially, four classes of Ab were categorized, based on the orientation of the RBD bound and whether the Ab blocks infectivity or binding to the cellular receptor, ACE2.... particular Abs and Nbs may not be unambiguously classified" As pointed out by the authors here, the Barnes classification of four classes of antibodies, classes 1-4, were not defined by the epitopes, but based on the orientation of RBD and whether the Ab competition with receptor binding. However, this classification system has been widely used to represent antibody epitopes, regardless of other factors, e.g. different angles of approach etc., which is one of the major reasons of the ambiguity, besides the "relatively small number of available structures" (line 83). Can the authors discuss more about the usage of the function-based classification system to represent epitopes?

We thank the reviewers for asking for a distinction of functional as compared with structural classification of epitopes. We have added the following explanation to the text (lines 80-86):

The functional classification of RBD epitopes, i.e. those that block infectivity of SARS-CoV-2, is valuable in identifying Abs likely to be of immediate therapeutic benefit during a rapidly spreading pandemic. The structure-based, function-agnostic, approach described here captures a broader set of RBD epitopes and is aimed primarily towards understanding the physico-chemical basis of epitope-paratope interactions. Such an understanding can enable predictions of antibody reactivities of new RBD variants based solely on RBD amino acid sequences.

2. In Figures 1a, 1c and supplementary Fig. 2, the bars are too narrow to distinguish and associate each bar with its respective residue.

The reviewer rightly expects to be able to visualize the direct correspondence of the plotted hits with the indicated amino acid sequence.

To clarify the correspondence of the amino acid and its number with the plotted hit, we have added an additional tick mark in the revised figures. Because of the compressed nature of the plot, we have only indicated the amino acid name and number of the even numbered ones. This is explained further in the Figure legends.

3. In the structural analyses, did the authors take account of structures with SARS-CoV-2 variants? Or only structures with wild-type spikes/RBDs were analyzed?

Of the curated structures, 59 of the 340 Ab and 4 of 83 Nb structures were determined with variant RBDs. Our analysis included all variants, but eliminating those variants from the analysis showed little difference in the residue/hit plots (e.g. Figure 1a, c). This information has been added to Supplementary Data 1 and Data 2 as an additional column indicating the mutations.

We have added a sentence clarifying this in the Methods on page 21, line 415-421.

Of the curated structures, 59 of the 340 Ab and 4 of 83 Nb structures were determined with variant RBDs. Our analysis included all variants, but eliminating those variants from the analysis showed little difference in the residue/hit plots. (See **Supplementary Data 1 and 2**).

4. In Figures 1a, 1c and Supplementary Fig. 2, many residues are missing, e.g. 417, 484, 486.

As indicated in Reviewer's point 2, above, for clarity, the odd-numbered amino acids were not labelled, but were present in the analysis. Residues E484 and F486 are labelled and plotted.

5. Page 17, line 329 "Although non-random factors may contribute to biases in the available database". The authors may want to expand discussions about biases introduced by the structural database analyzed here. For example, some antibodies may be frequently elicited in immune system but difficult to isolate owing to the stabilized bait used, or the difficulty of determining structures of some major classes of antibodies targeting cryptic sites on SARS-CoV-2 spike.

We appreciate the suggestion to expand the Discussion. Our analysis distinguishes the total of 340 antibodies from 83 nanobodies. We have added further Discussion of the various sources of the Ab/Nb in the database in lines 365 to 375.

Here we took advantage of currently available structures of complexes from a database of antibodies that derive largely from patients or vaccinees but also from mice and includes single chain antibodies (nanobodies) derived from immune or naïve libraries (see **Table 1**). Of course, analysis of structures compiled in any antibody database may be biased by a variety of factors including the biological source(s) of the antibodies (from natural infection or immunization; or from naïve or immune based libraries), whether they could be engineered effectively to produce adequate amounts of protein for X-ray or cryo-EM analysis, whether the proteins crystallized well, or how well they bind variant viral proteins. Despite such potentially confounding factors, several important consistent conclusions may be drawn: 1) common sites are recognized by both Abs or Nbs;

6. Page 9, lines 151-152 "The values of BSA range from 106 Å² (PDB 6XDG) to 1112 Å² (PDB 7N64) for 340 Abs and from 444 Å² (PDB 7JVB) to 1412 Å² (7D2Z) for the 83 Nbs." Did the authors analyzed the heavy chains only for mAbs? If yes, can the authors also provide BSA numbers for heavy + light chains?

We thank the reviewer for pointing out the need to parse the BSA for the H and L chains. We computed the BSA values for both the heavy and light chains separately in **Supplementary-data-abs.csv** files. We have added further clarification in the text, lines 165-168.

The total buried surface area (BSA) was also computed for each of 340 Abs (for H chain, L chain and H plus L chain) and for 83 Nbs as in **Supplementary Data 1** (for Abs) **and 2** (for Nbs). The values of BSA range from 64 Å² to 1112 Å² for Ab H, from 0 Å² to 912 Å² for Ab L, and between 264 and 1824 Å² for H plus L of the 340 Abs. BSA for the 83 Nb ranges from 437 Å² to 1412 Å².

7. When calculating BSA values of RBDs by antibodies, did the authors calculate the interface made by the RBD- N343 glycan?

No. We did not calculate the effect of glycosylation in the interactions. We mention this in the Discussion (lines 343 to 345).

This analysis is focused on the RBD alone and does not take into account potential contacts with the glycan moiety linked at N343 for which only 27 of 340 Ab make any contact.

8. Which antibody numbering system (Kabat? IMGT?) did the authors use in this study to assign residue numbers and CDR lengths in this study?

CDR designations were taken from the CoV-AbDab database, which follows the IMGT. The sequence designations of CDRH3 and CDRL3 were taken directly from the CoV-AbDab download. CDRH1, CDRH2, CDRL1, CDRL2 were identified by sequence alignment. We have added clarification. (p 20, lines 402 to 405)

(f) CDR designations were taken from the CoV-AbDab database, which follows the IMGT numbering system. The sequence designation of CDRH3 and CDRL3 were taken directly from the CoV-AbDab download. CDRH1, CDRH2, CDRL1, CDRL2 were identified by sequence alignment.

9. CDRH1 plays an exceptionally major role in binding ES16, which may be related to the big cluster A1S1. Is this observation related to the over-represented public class of mAbs encoded by IGHV3-53/3-66 [1,2]?

Yes. the A1S1 cluster is related to the mAbs encoded by IGHV3-53/3-66 as described by ^{1, 2, 3}. We tabulated the usage of particular human IGHV that covaried with particular IGHJ based on sequence, and found the most common sequences to be IGHV3-30 with IGHJ4. However, of the Ab structures, although IGHV3-30 was also preferentially found with IGHJ4, more structures were found with IGHV3-53 associated with either IGHJ4 or IGHJ6. This is now summarized in **Supplementary Fig. 6**, with further explanation on page 13, 14 lines 255 to 263.

Previous work identified the over-represented public class of mAbs encoded by IGHV3-53 and IGHV3-66 that neutralize the spike¹⁻³. We also investigated the V(D)J gene combinations representing those mAb structures (**Supplementary Fig.6a**). Among 6316 mAb sequences in the Cov-AbDab, the top three IGHV genes are 3-30, 1-69 and 3-53, and IGHJ genes are 4, 6 and 3. However, the top IGHV genes for the structural representatives are 3-53 and 3-58 and IGHJ gene 4, 6, and 3 combined (see blue heat map). A large cluster based on the gene combination similarity, GA1 (IGHV3-53/IGHJ6), as shown in **Supplementary Fig.6b**, has an ES set of (8,9,13,16,18,19) which is related to the cluster of A1S1 (**Figure 5c**). However, GA1 is a subset of the cluster A1S1 (17 vs 28 members).

New **Supplementary Fig. 6a and 6b**.

10. Page 12, lines 216-218: "Assigning a similarity threshold of 0.85 (see Methods) results in the identification of 33 distinct, non-overlapping, clusters for Abs, designated A1 to A33 (Supplementary Table 4a) and 10 distinct clusters for Nbs, N1 to N10 (Supplementary Table 4b)." But the antibodies shown in Supplementary Table 4a seem overlapped. For example, CV30, CC12-3 are in both A1 and A2. CR3022, C1C-A3, Asarnow-3D11 are in both A11 and A17. Some antibodies like REGN10987 are not found in any of these clusters.

Thank you for pointing this out. Our description of "non-overlapping" on the lines 216-218 was indeed incorrect. There is some small degree of overlap between the clusters because these clusters were defined by "similarity=0.85 of ES". (line 235).

We have deleted "distinct" from line 233 and 234, and "non-overlapping" from line 235.

11. Table 6 cited in the main text should be Table 3?

Yes. Table 6 should be **Table 3**. We have corrected this.

In Figure 6 and Table 3, S494P and E484K are listed as Alpha mutations, and K417N in Delta. But normally these mutations are not considered as signature mutations in these variants. In addition, Some Omicron subvariants can contain G339H and T478R, in addition to G339D and T478K listed here.

Yes, thank you. You are right. S494P and E484K are not the signature mutations in the Alpha variant, nor is K417N in Delta, but they were detected in some but not all variants. We have corrected Table 3 by labelling these in Alpha and Delta with an asterisk, *. We also note this in the Figure and Table legends and list the Omicron variants in additional columns. Thus, G339H and T478R are added in Table 3a where, for example, "X" means "D, H" for G339X and "K, R" for T478X.

12. Figure 6d: What are differences of pink and red residues? Why are some of the ESs highlighted?

Here, we originally meant the red colored residues to emphasize those that have multiple amino acid substitutions (subvariants of Omicron), and the salmon color indicated a single mutation of the residue. We have simplified this now and have changed all Omicron mutations to a uniform color as “salmon”.

13. Page 15, line 278: R486 should be F486.

Thank you for picking this up. It has been corrected. (now, line 305)

14. Page 15, line 287: 7L7E contains both tixagevimab and cligavimab.

You are right, this has been corrected. (now line 313)

Reviewer #2 (Remarks to the Author):

In this manuscript, the authors have utilized a computational clustering method and available crystallographic structures to identify 23 unique epitomic sites on the receptor binding domain (RBD) of the SARS-CoV-2 virus, the causative agent of the COVID-19 pandemic. This study is both intriguing and relevant, as it offers potential insights for the development of small molecules/proteins/peptides capable of neutralizing concerning variants that exhibit mutations in the RBD. Additionally, I appreciate the authors' examination of the distinctions between antibody (Abs) and nanobody (Nbs) epitope sites, a comparison that has received limited attention in prior research studies. Previous studies, one example listed here: <https://www.frontiersin.org/articles/10.3389/fimmu.2021.691715/full> have focused on a much smaller subset of epitope sites, and as the author discuss in their introduction, not all Abs can be classified accurately within these “generic” classification schemes. I think the level of detail in 23 epitope sites and the 33 distant, non-overlapping clusters for Abs, will be useful in characterizing newly discovered Abs with less potential for immune escape, as well as potentially targeting the RBD with small molecule/peptide inhibitors at these epitope sites that are more resistant to immune escape. The analysis conducted by the author also offers insights into the immune escape of other commercial antibodies, including Evushield. This serves as a valuable example of how this research can provide a rationale for understanding immune escape mechanisms, particularly as new variants of SARS-CoV-2 continue to emerge. However, there are mistakes in the text, tables and figures. Therefore, I suggest that this paper be considered for publication in Communications Biology once the following issues have been addressed:

1. The authors need to check the text, figures, and tables as they don't match in many places.

Thanks for noticing these mismatched figures and tables with the text. We apologize for any confusion that these problems may have generated, and have thoroughly examined the correspondence of text, figures, and tables, and trust that these are now okay. Also, we thank the reviewer for informing us of the Deshpande et al classification and have referred to it in the text (ref. 22).

a. Line 177, the authors say “each ES surface area or footprint is illustrated by a color map of the RBD surface (Figure 2d). I think the authors mean Figure 2e? Please check.

Yes. It is Figure 2e, Corrected. (now line 193)

b. Line 242, I do not see Figure 5d in the files supplied for review? It is included in Fig.5 description but there is no corresponding 5d figure.

We inadvertently omitted the reference to Figure 5d in the original submission. This has been corrected.(now line 268)

c. There is no Table 6a, maybe it is Table 3a

Yes. It is Table 3a, and the call out has been corrected. (now Line 276)

d. ES2 is not present in Figure 6a, maybe it is 6d

Yes, it is Figure 6d. This has been corrected.

e. line 275: there is not supplementary figure 5a

It is **Supplementary Figure 4a** and has been corrected. (now line 300)

f. Line 250 and line 252, authors list table 6a and table 6b. This is now table 3a and table 3b in their supplied files. Please fix throughout the manuscript where it is referred to as table

All "**Table 6**" are changed to "**Table 3**". We again apologize for the confusion.

g. Line 275, there is no supp. Figure 5a? I think the authors mean Figure 4a? Same with line 282.

These are now properly referred to as **Supplementary Figure 4a, not 5a**.

2. Can the authors include a little bit in the introduction about new omicron subvariants such as XBB.1.5 and how all FDA approved Abs are no longer effective.

Thank you for the suggestion. This is, of course, a moving target. We have augmented the reference to Ab and Omicron variants at the end of the Results. Lines, 319-326.

In particular, Omicron variant XBB.1.5 (see **Table 3b**), harbors mutations that reduce efficient recognition by the Ab products (bamlanivimab plus etesevimab, casirivimab plus imdevimab, sotrovimab, and bebtelovimab), which are no longer authorized for use in the United States <https://www.covid19treatmentguidelines.nih.gov/therapies/antivirals-including-antibody-products/anti-sars-cov-2-monoclonal-antibodies/#:~:text=Four%20anti%2DSARS%2DCoV%2D,mild%20to%20moderate%20COVID%2D19.>

3. How did the authors decide on a cut-off distance of 5.0 Å for their Ab-Ag interface contacts?

This is an important point. We have explained the rationale for cut-off distances in the text (lines 128-138), and substantiate our approach empirically in the Discussion with reference to **Supplementary Figure 5**.

For the hydrogen bond interactions, the distance is usually cut-off about 3.8-4.0 angstrom, but for non-bonded interaction or van der Waals interaction, it could be up to 5-8 angstroms. Generally, distances within 4-6 angstroms (Å) are considered indicative of direct contacts between interacting proteins. But for computational approaches it may use cut-offs range from 5 to 10 Å due to the dynamics feature of proteins. Vangone and Bonvin ⁴ studied the correlation of the contact distance and the binding affinity, and found the approximate distance range is between 4.0 to 5.5 Å. Vilorio et al.,⁵ determined an optimal distance cut-off for contact-based protein networks of 5.0 Å. Krawzyk et al. ⁶ used 4.5 Å to define as epitope contacts. We adopted the contact distance cut-off at 5.0 Å for our Ab-Ag interaction, based on comparison of different cut-offs from 4.0 to 5.5 Å (see Methods).

(351-356)

Our findings and definition of 23 ES are not dependent on the distance cut-off parameter in computing the Ag-Ab contacts. Although the total numbers of contacts may increase slightly with a longer distance cut-off, the percentage of contacts in each ES bin are almost the same (as shown in **Supplementary Figure 5** with the distance cut-offs at 4.0 Å (gray), 4.5 Å (blue), 5.0 Å (red) and 5.5 Å (green) respectively (H-chain only).

Supplementary Figure 5.

4. Table 3 (what authors have as Table 6 in the text) contains spelling mistake “SERS-cov-2”. Please fix.

Thank you, corrected.

5. Line 259, authors state mutations in ES3,6,9,14,15, and 23 have not yet been reported. Could the authors create a separate figure with these epitopes highlighted and their corresponding residues? I think this will be helpful in showing where Abs or Nbs or other inhibitors could be targeted with less chance for immune escape by omicron.

Thank you for the suggestion. We have added **Figure 6e** to illustrate this point. (lines 286-287).

These ES for which mutations have not yet been reported are illustrated in **Figure 6e**.

6. Discussion: line 337, why do the authors recommend ES11, 13, 16, and 18 as sites to incorporate into peptide-based immunogens? Please provide a clarifying sentence based on your data why you recommend these sites.

ES11, 13, 16, 18, and 20 have relatively higher percentage of interaction with Ab (see Table 2). They are indicated in boldface in the Table and are now explained better in the text. (lines 379 to 381).

Our analysis suggests that several regions of the RBD, that are recognized by a higher proportion of Ab may be particularly important to incorporate into peptide-based immunogens (such as ES11, 13, 16, 18, and 20) and that further generation vaccines might pay particular attention to new viral variants that affect these sites.

7. Methods: line 368, please fix language “which users can make inquire”. Maybe change to “in which users can inquire about a particular ES combination”.

Thank you for pointing this out. The English has been improved.

We also provide a program by which users may inquire about particular ES combinations, PDB ID, Ab name, or Class 1-4 designation, with a given similarity threshold.

8. Figure 4d: This figure is only for CDR3; authors should also show figures for CDR1 and CDR2.

Thank you for the suggestion. We have added **Figure 4e** for CDR2 and **Figure 4f** for CDR1. These are also referred to in the text, lines 223-225.

The usage of CDR3, CDR2 and CDR1 amino acids is plotted in **Figure 4d, 4e and 4f** respectively.

9. Line 191, in addition to Figure 3b, authors should also mention Figure 5C to show the CDR1, CDR2 and CDR3 regions.

We have added reference to the illustration of Figure 5c (line 212 - 214)

Thus, for both Ab H chains and Nbs, CDR3 contributes the greater proportion of those residues that interact with the RBD, reflecting a major role for CDR3 in RBD recognition. (Illustrations of CDR1, CDR2, and CDR3 contacts are shown in **Figure 5c**).

10. Supplementary information: Authors should change “Excel files for Tables 2a and 2b are attached separately” to “CSV files (with names of the files) for Tables 2a and 2b are attached separately.”

Yes. These are referred to as CSV files. Now they are changed to **Supplementary Data 1 and 2**.

11. The article would have been more appealing if the authors had provided suggestions for antibodies, peptide inhibitors, mini proteins, protein mimics, or decoy receptors (such as sACE2) that could target a majority of the epitopic sites (from the 23 identified) in the currently circulating Omicron variants and demonstrate their ability to neutralize them through in vitro and in vivo efficacy data.

We appreciate the reviewer’s useful suggestion. This is clearly an extensive project in itself that we are pursuing.

Reviewer #3 (Remarks to the Author):

The work of Jiang and colleagues deals with the investigation of the interaction between the receptor binding domain of SARS-CoV2 spike protein and a set of antibodies and nanobodies using structural data. In particular, studying the number of intermolecular contacts, the authors identified different epitopic sites on the RBD and cluster the antibodies in classes.

Overall the theme of understanding the interplay between antibodies and viral spike protein is important both from a theoretical and practical point of view. This work provides a detailed computational analysis, however, there are some points that in my opinion must be addressed before considering it for publication.

1) Presented results are obtained on the basis of the collected dataset, which comprises the most recent deposited structures. However, it is not clear to me what is the major progress with respect to other works already present in literature (see for instance Di Rienzo et al. "Dynamical changes of SARS-CoV-2 spike variants in the highly immunogenic regions impact the viral antibodies escaping", *Proteins: Structure, Function, and Bioinformatics*, <https://doi.org/10.1002/prot.26497> or Deshpande et al. 'Epitope Classification and RBD Binding Properties of Neutralizing Antibodies Against SARS-CoV-2 Variants of Concern. *Front Immunol.* 2021 Jun 4;12:691715. doi: 10.3389/fimmu.2021.691715).

The authors should at least better discuss similarities and differences between their work and what is now present in literature.

We appreciate the reviewer’s points to other works present in the literature. We have made reference to DiRienzo and Deshpande and have added further Discussion. (lines 76-80)

Epitopic analysis was further extended by the definition of seven “communities” of Abs that bind to the RBD surface⁷. Recent analysis of anti-RBD Ab and Nb as well as molecular dynamics analysis in the context of evolving escape mutations has taken advantage of these earlier classification schemes⁸⁻¹³. Others have analyzed a number of anti-spike Nb in terms of their affinity and neutralization capacity¹⁴.

2) SARS-CoV-2 spike protein is heavily glycosylated (see for instance Grant et al. "Analysis of the SARS-CoV-2 spike protein glycan shield: implications for immune recognition." *Sci Rep.* 2020 Sep 14;10(1):14991.)

It is not clear if the authors consider this factor in their assessment of the epitopic sites.

We did not discuss the effects of glycosylation on interactions with Ab. We addressed this above for Reviewer #1. We focus on the RBD domain which has only one glycosylation site (N343). We found that only 27 of 340 antibodies have close contacts with N343 regardless of the presence of the glycan.

We have added a comment about this in the discussion. (lines 345-347)

This analysis is focused on the RBD alone and does not take into account potential contacts with the glycan moiety linked at N343. Only 27 of 340 Ab make any contact with residue N343.

3) Found results rely on a particular choice of cut-off distance (5 Å) for the definition of contacts and of the clustering method. Do the authors explore the robustness of their findings upon changing such parameters?

Reviewer #2 in point 3 above raised the same question. We repeat our response here:

This is an important point. We have explained the rationale for cut-off distances in the text (lines 128-138) and substantiate our approach empirically in the Discussion with reference to **Supplementary Figure 5**.

(128-138)

For the hydrogen bond interactions, the distance is usually cut-off about 3.8-4.0 angstrom, but for non-bonded interaction or van der Waals interaction, it could be up to 5-8 angstroms. Generally, distances within 4-6 angstroms (Å) are considered indicative of direct contacts between interacting proteins. But for computational approaches it may use cut-offs range from 5 to 10 Å due to the dynamics feature of proteins. Vangone and Bonvin ⁴ studied the correlation of the contact distance and the binding affinity, and found the approximate distance range is between 4.0 to 5.5 Å. Vioria et al.,⁵ determined an optimal distance cut-off for contact-based protein networks of 5.0 Å. Krawzyk et al. ⁶ used 4.5 Å to define as epitope contacts. We adopted the contact distance cut-off at 5.0 Å for our Ab-Ag interaction, based on comparison of different cut-offs from 4.0 to 5.5 Å (see Methods).

(351-356)

Our findings and definition of 23 ES are not dependent on the distance cut-off parameter in computing the Ag-Ab contacts. Although the total numbers of contacts may increase slightly with a longer distance cut-off, the percentage of contacts in each ES bin are almost the same (as shown in **Supplementary Figure 5** with the distance cut-offs at 4.0 Å (gray), 4.5 Å (blue), 5.0 Å (red) and 5.5 Å (green) respectively (H-chain only).

4) Please provide more details about the dataset. It is not clear how/if redundancy was taken into account, what resolution for the structure was considered, how missing residues were fixed, etc.

This is now detailed in Methods, lines 392-430.

Using the Cov-AbDab list as of 12/20/2022, we downloaded all complexes of Ab/spike or Ab/RBD structures determined by X-ray crystallography or cryoEM from the PDB. The total number of downloaded PDB entries is about 595 (see the “Related Structures” column in **Supplementary Data 1**). We manually curates this structure dataset as follows: (a) We removed the structures with resolution worse than 5.0 Å; (b) when the same Ab appeared in two or more different structures, we selected the one of highest resolution; (c) if an Ab had both X-ray and cryoEM structures, we removed the one that was of worse resolution, regardless of the method; (d) since the spike is a trimer, if two or more of the same Ab bound to different chains of the same spike, we considered only one complex with this Ab; (e)

in the case of bispecific or bivalent Ab or multiple Ab/Nb in the same PDB structure, we evaluated two or more such “unique” Ab/RBD complexes. (f) CDR designations were taken from the CoV-AbDab database, which follows the IMGT numbering system. The sequence designation of CDRH3 and CDRL3 were taken directly from the CoV-AbDab download. CDRH1, CDRH2, CDRL1, CDRL2 were identified by sequence alignment.

This procedure resulted in 340 unique “non-redundant” neutralizing antibodies from 595 PDB entries (see **Supplementary Data 1**). We performed the same analysis for Nb (from a list of 124 structures reduced to 83 “non-redundant” Nb, see **Supplementary Data 2**). We did not rectify any missing residues or atoms and kept the original model from the downloaded PDB structures. We used the CNS-1.3 script, “contact.inp” with the distance cut-off at 5.0 angstroms to compute the interacting contacts between Abs and RBD for these 340 structures. If two or more residues from an Ab contacted identical residues on the RBD within 5.0 angstroms, we list them as multiple contacts. This procedure produced the “raw” epitope-paratope contact dataset (see github.com/jiangj-niaid/RBD-SARS2/340absH-contact-dis-0207.txt) for the following analysis step. Of the curated structures, 59 of the 340 Ab and 4 of 83 Nb structures were determined with variant RBDs. Our analysis included all variants, but eliminating those variants from the analysis showed little difference in the residue/hit plots. (See **Supplementary Data 1&2**).

Buried surface area (BSA) is an important structural character and a quantitative measurement of interaction at the interface. BSA values correlate to the sum of individual contacts and directly link to the binding affinity or neutralizing potency. We also calculate the BSA values (using PISA/CCP4 program) for each chain (H and L) of an antibody/nanobody.

Based on these curations and evaluations, we created the annotation files for each Abs and Nbs; see **Supplementary Data 1 and Data 2**. [“suppl-Table2A-abs.csv” file contains the following data items: Name-of-Abs, PDB id, epitope-chain-id, paratope-chain-ids, R/S (RBD alone or spike), X/E (X-ray or cryoEM), Resolution, BSA H-chain, BSA L-chain, ES H-chain, ES L-chain, Related Structures, Variants-mark. “suppl-Table2B-nbs.csv” contains the following data items: Name-of-Nbs, PDB id, epitope-chain-id, paratope-chain-id, R/S (RBD alone or spike), X/E (X-ray or cryoEM), Resolution, BSA, ES, Related Structures, Variants-mark. (see github.com/jiangj-niaid/RBD-SARS2/)].

5) Please provide also the key details on the used algorithms (not only the reference to the software)

We have added more details of the software package descriptions and algorithms in the Methods section (lines 434 to 446), and we have also added a flowchart of the EPI software package, as shown in **Supplementary Fig.7**, to highlight the key details of the procedures. (Lines 439 to 449)

Supplementary Figure 7 shows the flowchart of this package. We downloaded the sequences, including the extracted CDR sequences from CovAbDab, also downloaded all structures of Ab/Nb in complex with spike or RBD from the PDB and deduced a “non-redundant” structural dataset (see “Datasets” above). The Input files and parameters include: (a) A list of PDB IDs and names of Ab/Nb along with the epitope chain and paratope chain designations. (b) Predefined ES residue range (in **Table 2a**). (c) Predefined CDR loops. (d) A list of known variants and mutations. (e) The contact distance cut-off (default is 5.0 Å). The program will generate “EPI contact datasets” according to the input files and pdbid list. Separate sub-dataset for Ab H-chain, L-chain, or Nb may be designated. Based on this EPI dataset, the analysis scripts then perform a structural alignment and evaluate the statistics of the CDR loops and amino acid usage. Additional variants and mutants may be detected.

Supplementary Figure 7.

REFERENCES FOR “Response to Reviewers”

- 1 Yuan, M. *et al.* Structural basis of a shared antibody response to SARS-CoV-2. *Science* **369**, 1119-1123 (2020). <https://doi.org:10.1126/science.abd2321>
- 2 Zhang, Q. *et al.* Potent and protective IGHV3-53/3-66 public antibodies and their shared escape mutant on the spike of SARS-CoV-2. *Nat Commun* **12**, 4210 (2021). <https://doi.org:10.1038/s41467-021-24514-w>
- 3 Robinson, S. A. *et al.* Epitope profiling using computational structural modelling demonstrated on coronavirus-binding antibodies. *PLoS Comput Biol* **17**, e1009675 (2021). <https://doi.org:10.1371/journal.pcbi.1009675>
- 4 Vangone, A. & Bonvin, A. M. Contacts-based prediction of binding affinity in protein-protein complexes. *Elife* **4**, e07454 (2015). <https://doi.org:10.7554/eLife.07454>
- 5 Salamanca Vilorio, J., Allega, M. F., Lambrugh, M. & Papaleo, E. An optimal distance cutoff for contact-based Protein Structure Networks using side-chain centers of mass. *Sci Rep* **7**, 2838 (2017). <https://doi.org:10.1038/s41598-017-01498-6>
- 6 Krawczyk, K., Liu, X., Baker, T., Shi, J. & Deane, C. M. Improving B-cell epitope prediction and its application to global antibody-antigen docking. *Bioinformatics* **30**, 2288-2294 (2014). <https://doi.org:10.1093/bioinformatics/btu190>
- 7 Hastie, K. M. *et al.* Defining variant-resistant epitopes targeted by SARS-CoV-2 antibodies: A global consortium study. *Science* **374**, 472-478 (2021). <https://doi.org:10.1126/science.abh2315>
- 8 Greaney, A. J. *et al.* The SARS-CoV-2 Delta variant induces an antibody response largely focused on class 1 and 2 antibody epitopes. *PLoS Pathog* **18**, e1010592 (2022). <https://doi.org:10.1371/journal.ppat.1010592>
- 9 Greaney, A. J. *et al.* Mapping mutations to the SARS-CoV-2 RBD that escape binding by different classes of antibodies. *Nat Commun* **12**, 4196 (2021). <https://doi.org:10.1038/s41467-021-24435-8>
- 10 Starr, T. N. *et al.* SARS-CoV-2 RBD antibodies that maximize breadth and resistance to escape. *Nature* **597**, 97-102 (2021). <https://doi.org:10.1038/s41586-021-03807-6>
- 11 Lubin, J. H. *et al.* Structural models of SARS-CoV-2 Omicron variant in complex with ACE2 receptor or antibodies suggest altered binding interfaces. *bioRxiv* (2021). <https://doi.org:10.1101/2021.12.12.472313>
- 12 Deshpande, A., Harris, B. D., Martinez-Sobrido, L., Kobie, J. J. & Walter, M. R. Epitope Classification and RBD Binding Properties of Neutralizing Antibodies Against SARS-CoV-2 Variants of Concern. *Front Immunol* **12**, 691715 (2021). <https://doi.org:10.3389/fimmu.2021.691715>
- 13 Di Rienzo, L. *et al.* Dynamical changes of SARS-CoV-2 spike variants in the highly immunogenic regions impact the viral

antibodies escaping. *Proteins* (2023). <https://doi.org:10.1002/prot.26497>

Rossotti, M. A. *et al.* Arsenal of nanobodies shows broad-spectrum neutralization against SARS-CoV-2 variants of concern in vitro and in vivo in hamster models. *Commun Biol* 5, 933 (2022). <https://doi.org:10.1038/s42003-022-03866-z>

REVIEWERS' COMMENTS:

Reviewer #1 (Remarks to the Author):

Figure 6. Both signature and non-signature mutations are shown for each variant - it would be great to add a note to show which of these are non-signature mutations.

Reviewer #3 (Remarks to the Author):

The authors addressed my comments. I only suggest to add the specifics on how contacts are calculated to ensure reproducibility of the work (are distances computed between carbon alpha or closest atoms?).

Reviewer #4 (Remarks to the Author):

The authors have addressed all the questions and concerns.

RESPONSE TO REVIEWERS

REVIEWERS' COMMENTS:

Reviewer #1 (Remarks to the Author):

Figure 6. Both signature and non-signature mutations are shown for each variant - it would be great to add a note to show which of these are non-signature mutations.

We agree. We have added a note to the legend to Figure 6 concerning the non-signature mutations.

Reviewer #3 (Remarks to the Author):

The authors addressed my comments. I only suggest to add the specifics on how contacts are calculated to ensure reproducibility of the work (are distances computed between carbon alpha or closest atoms?).

We have added the details in the methods that these are the closest atoms.

Reviewer #4 (Remarks to the Author):

The authors have addressed all the questions and concerns.

We appreciate the reviewers' suggestions and concerns.